# An Accelerated Proximal Algorithm for Regularized Nonconvex and Nonsmooth Bi-level Optimization

## Abstract

Many important machine learning applications involve regularized nonconvex bi-level optimization. However, the existing gradient-based bi-level optimization algorithms cannot handle nonconvex or nonsmooth regularizers, and they suffer from a high computation complexity in nonconvex bi-level optimization. In this work, we study a proximal gradient-type algorithm that adopts the approximate implicit differentiation (AID) scheme for nonconvex bi-level optimization with possibly nonconvex and nonsmooth regularizers. In particular, the algorithm applies the Nesterov's momentum to accelerate the computation of the implicit gradient involved in AID. We provide a comprehensive analysis of the global convergence properties of this algorithm through identifying its intrinsic potential function. In particular, we formally establish the convergence of the model parameters to a critical point of the bi-level problem, and obtain an improved computation complexity $\widetilde{\mathcal{O}}(\kappa^{3.5}\epsilon^{-2})$ over the state-of-the-art result. Moreover, we analyze the asymptotic convergence rates of this algorithm under a class of local nonconvex geometries characterized by a Łojasiewicz-type gradient inequality. Experiment on hyper-parameter optimization demonstrates the effectiveness of our algorithm.

## 1 Introduction

Bi-level optimization has become an important and popular optimization framework that covers a variety of emerging machine learning applications, e.g., meta-learning (Franceschi et al., 2018; Bertinetto et al., 2018; Rajeswaran et al., 2019; Ji et al., 2020a), hyperparameter optimization (Franceschi et al., 2018; Shaban et al., 2019; Feurer & Hutter, 2019), reinforcement learning (Konda & Tsitsiklis, 2000; Hong et al., 2020), etc. A standard formulation of bi-level optimization takes the following form.

$$\min_{x \in \mathbb{R}^d} f(x, y^*(x)), \quad \text{where} \quad y^*(x) \in \arg\min_{y \in \mathbb{R}^p} g(x, y),$$

where the upper- and lower-level objective functions $f$ and $g$ are both jointly continuously differentiable. To elaborate, bi-level optimization aims to minimize the upper-level compositional objective function $f(x, y^*(x))$, in which $y^*(x)$ is the minimizer of the lower-level objective function $g(x, y)$.

Solving the above bi-level optimization problem is highly non-trivial as the problem involves two nested minimization problems. In the existing literature, many algorithms have been developed for bi-level optimization. In the early works, Hansen et al. (1992); Shi et al. (2005); Moore (2010) reformulated the bi-level problem into a single-level problem with constraints on the optimality conditions of the lower-level problem, yet this reformulation involves a large number of constraints that are hard to address in practice. More recently, gradient-based bi-level optimization algorithms have been developed, which leverage either the approximate implicit differentiation (AID) scheme (Domke, 2012; Pedregosa, 2016; Gould et al., 2016; Liao et al., 2018; Ghadimi & Wang, 2018; Grazzi et al., 2020; Lorraine et al., 2020) or the iterative differentiation (ITD) scheme (Domke, 2012; Maclaurin et al., 2015; Franceschi et al., 2017; 2018; Shaban et al., 2019; Grazzi et al., 2020) to estimate the gradient of the upper-level function. In particular, the AID scheme is more popular due to its simplicity and computation efficiency. Specifically, bi-level optimization algorithm with

AID (referred to as BiO-AID) has been analyzed for (strongly)-convex upper- and lower-level functions (Liu et al., 2020), which do not cover bi-level problems in modern machine learning that usually involve nonconvex upper-level objective functions. On the other hand, recent studies have analyzed the convergence of BiO-AID with nonconvex upper-level function and strongly convex lower-level function, and established the convergence of a certain type of gradient norm to zero (Ji et al., 2021; Ghadimi & Wang, 2018; Hong et al., 2020).

However, the existing gradient-based nonconvex bi-level optimization algorithms have limitations in several perspectives. First, they are not applicable to bi-level problems that involve possibly nonsmooth and nonconvex regularizers. For example, in the application of data hyper-cleaning, one can improve the learning performance by adding a nonsmooth and nonconvex regularizer to push the weights of the clean samples towards 1 while push those of the contaminated samples towards 0 (see Section 6 for more details). Second, the convergence guarantees of these algorithms typically ensure a weak gradient norm convergence, which does not necessarily imply the desired convergence of the model parameters. Furthermore, these algorithms suffer from a high computation complexity in nonconvex bi-level optimization. **The overarching goal of this work** is to develop an efficient and convergent proximal-type algorithm for solving regularized nonconvex and nonsmooth bi-level optimization problems and address the above important issues. We summarize our contributions as follows.

## 1.1   Our Contributions

We propose a proximal BiO-AIDm algorithm (see Algorithm 1) and study its convergence properties. This algorithm is a proximal variant of the BiO-AID algorithm for solving the following class of regularized nonsmooth and nonconvex bi-level optimization problems.

$$\min_{x \in \mathbb{R}^d} f(x, y^*(x)) + h(x), \text{where } y^*(x) = \arg\min_{y \in \mathbb{R}^p} g(x, y),$$

where the upper-level objective function $f$ is nonconvex, the lower-level objective function $g$ is strongly convex for any fixed $x$, and the regularizer $h$ is possibly nonsmooth and nonconvex. In particular, our algorithm applies the Nesterov's momentum to accelerate the computation of the implicit gradient involved in the AID scheme.

We first analyze the global (non-asymptotic) convergence properties of proximal BiO-AIDm under standard Lipschitz and smoothness assumptions on the objective functions. The key to our analysis is to show that proximal BiO-AID admits an intrinsic potential function $H(x_k, y_k)$ that takes the form

$$H(x, y') := \Phi(x) + h(x) + \frac{7}{8}\|y^{(T)}(x, y') - y^*(x)\|^2,$$

where $y^{(T)}(x, y')$ is obtained by applying the Nesterov's accelerated gradient descent to minimize $g(x, \cdot)$ with initial point $y'$ for $T$ iterations. In particular, we prove that such a potential function is monotonically decreasing along the optimization trajectory, i.e., $H(x_{k+1}, y_{k+1}) < H(x_k, y_k)$, which implies that proximal BiO-AIDm can be viewed as a descent-type algorithm and is numerically stable. Based on this property, we formally prove that every limit point of the model parameter trajectory $\{x_k\}_k$ generated by proximal BiO-AIDm is a critical point of the regularized bi-level problem. Furthermore, when the regularizer is convex, we show that proximal BiO-AIDm requires a computation complexity of $\widetilde{\mathcal{O}}(\kappa^{3.5}\epsilon^{-2})$ (number of gradient, Hessian-vector product and proximal evaluations) for achieving a critical point $x$ that satisfies $\|G(x)\| \le \epsilon$, where $\kappa$ denotes the problem condition number and $G(x)$ denotes the proximal gradient mapping. As shown in Table 1, this is the first global convergence and complexity result of proximal BiO-AIDm in regularized nonsmooth and nonconvex bi-level optimization, and it improves the state-of-the-art complexity of BiO-AID (for smooth nonconvex bi-level optimization) by a factor of $\widetilde{\mathcal{O}}(\sqrt{\kappa})$.

Besides investigating the global convergence properties, we further establish the asymptotic function value convergence rates of proximal BiO-AIDm under a local Łojasiewicz-type nonconvex geometry, which covers a broad spectrum of local nonconvex geometries. Specifically, we characterize the asymptotic convergence rates of proximal BiO-AIDm in the full spectrum of the Łojasiewicz geometry parameter $\theta$. We prove that as the local geometry becomes sharper (i.e., with a larger $\theta$), the asymptotic convergence rate of proximal

BiO-AIDm boosts from sublinear convergence to superlinear convergence. The proof of these local asymptoic convergence rates requires proving two properties that have not been proved in the existing literature to our knowledge. The major property is that the aforementioned potential function $H$ is decreasing. Another novel property is the Lipschitz property of $y^{(T)}$, which is challenging to prove due to momentum acceleration.

Table 1: List of existing complexity results for bi-level algorithms. ($\checkmark$ in the columns "non-smooth" and "momentum accelerated" respectively means the objective function is non-smooth and the algorithm has momentum acceleration, and $\times$ means the opposite.)

| | upper level function | lower level function | non-smooth | momentum accelerated | computation complexity |
|---|---|---|---|---|---|
| Ghadimi & Wang (2018) | nonconvex | strongly convex | $\times$ | $\times$ | $\mathcal{O}(\kappa^4\epsilon^{-2})$ |
| Ji et al. (2021) | nonconvex | strongly convex | $\times$ | $\times$ | $\mathcal{O}(\kappa^4\epsilon^{-2})$ |
| **This work** | nonconvex | strongly convex | $\checkmark$ | $\checkmark$ | $\mathcal{O}(\kappa^{3.5}(\ln\kappa)\epsilon^{-2})$ |

## 1.2 Related Work

**Bi-level Optimization Algorithms.** Bi-level optimization has been studied for decades (Bracken & McGill, 1973), and various types of bi-level algorithms have been proposed, including but not limited to single-level penalized methods (Shi et al., 2005; Moore, 2010), and gradient-based methods via AID or ITD-based hypergradient estimation (Domke, 2012; Pedregosa, 2016; Franceschi et al., 2018; Ghadimi & Wang, 2018; Hong et al., 2020; Liu et al., 2020; Li et al., 2020; Grazzi et al., 2020; Ji et al., 2021; Lorraine et al., 2020; Ji & Liang, 2021). In particular, (Ghadimi & Wang, 2018; Hong et al., 2020; Ji et al., 2021; Yang et al., 2021; Chen et al., 2021a; Guo & Yang, 2021) characterized the complexity analysis for their proposed methods for bi-level optimization problem under different types of loss geometries. (Ji & Liang, 2021) studied the lower complexity bounds for bi-level optimization under (strongly) convex geometry and proposed a nearly-optimal accelerated algorithm. All the existing analysis of nonconvex bi-level optimization algorithms focuses on the gradient norm convergence. In this paper, we formally establish the parameter and function value convergence of proximal BiO-AID in regularized nonconvex and nonsmooth bi-level optimization.

**Applications of Bi-level Optimization.** Bi-level optimization has been widely applied to meta-learning (Snell et al., 2017; Franceschi et al., 2018; Rajeswaran et al., 2019; Zügner & Günnemann, 2019; Ji et al., 2020b; Ji, 2021), hyperparameter optimization (Franceschi et al., 2017; Shaban et al., 2019), reinforcement learning (Konda & Tsitsiklis, 2000; Hong et al., 2020), and data poisoning (Mehra et al., 2020). For example, Snell et al. (2017) reformulated the meta-learning objective function under a shared embedding model into a bi-level optimization problem. Rajeswaran et al. (2019) proposed a bi-level optimizer named iMAML as an efficient variant of model-agnostic meta-learning (MAML) (Finn et al., 2017), and analyzed the convergence of iMAML under the strongly convex inner-loop loss. Fallah et al. (2020) characterized the convergence of MAML and first-order MAML under nonconvex loss functions. Ji et al. (2020a) studied the convergence behaviors of almost no inner loop (ANIL) (Raghu et al., 2019) under different inner-loop loss geometries of the MAML objective function. Recently Mehra et al. (2020) devised bi-level optimization based data poisoning attacks on certifiably robust classifiers.

**Nonconvex Kurdyka-Łojasiewicz Geometry.** A broad class of regular functions has been shown to satisfy the local nonconvex KŁ geometry (Bolte et al., 2007), which affects the asymptotic convergence rates of gradient-based optimization algorithms. The KŁ geometry has been exploited to study the convergence of various first-order algorithms for solving minimization problems, including gradient descent (Attouch & Bolte, 2009), alternating gradient descent (Bolte et al., 2014), distributed gradient descent (Zhou et al., 2016; 2018a), accelerated gradient descent (Li et al., 2017). It has also been exploited to study the convergence of second-order algorithms such as Newton's method (Noll & Rondepierre, 2013; Frankel et al., 2015) and cubic regularization method (Zhou et al., 2018b).

## 2 Problem Formulation and Preliminaries

In this paper, we consider the following regularized nonconvex bi-level optimization problem:

$$\min_{x \in \mathbb{R}^d} f(x, y^*(x)) + h(x), \text{where } y^*(x) \in \arg\min_{y \in \mathbb{R}^p} g(x, y), \tag{P}$$

where both the upper-level objective function $f$ and the lower-level objective function $g$ are jointly continuously differentiable, and the regularizer $h$ is possibly nonsmooth and nonconvex. We note that adding a regularizer to the bi-level optimization problem allows us to impose desired structures on the solution, and this is important for many machine learning applications. For example, in the application of data hyper-cleaning (see the experiment in Section 6 for more details), one aims to improve the learning performance by adding a regularizer to push the weights of the clean samples towards 1 while push the weights of the contaminated samples towards 0. Such a regularizer often takes a nonsmooth and nonconvex form.

To simplify the notation, throughout the paper we define the function $\Phi(x) := f(x, y^*(x))$. We also adopt the following standard assumptions regarding the regularized bi-level optimization problem (P).

**Assumption 1.** *The functions in the regularized bi-level optimization problem (P) satisfy:*

*1. Function $g(x, \cdot)$ is $\mu$-strongly convex for all $x$ and function $\Phi(x) = f(x, y^*(x))$ is nonconvex;*

*2. Function $h$ is proper and lower-semicontinuous (possibly nonsmooth and nonconvex);*

*3. Function $(\Phi + h)(x)$ is bounded below and has bounded sub-level sets.*

In Assumption 1, the regularizer $h$ can be any nonsmooth and nonconvex function so long as it is closed. This covers most of the regularizers that we use in practice. In addition to Assumption 1, we also impose the following Lipschitz continuity and smoothness conditions on the objective functions, which are widely considered in the existing literature (Ghadimi & Wang, 2018; Ji et al., 2020a). In the following assumption, we denote $z := (x, y)$.

**Assumption 2.** *The functions $f(z)$ and $g(z)$ in the bi-level problem (P) satisfy:*

*1. Function $f(z)$ is $M$-Lipschitz. Gradients $\nabla f(z)$ and $\nabla g(z)$ are $L_f$-Lipschitz and $L_g$-Lipschitz respectively*

*2. Jacobian $\nabla_x \nabla_y g(z)$ and Hessian $\nabla_y^2 g(z)$ are $\tau$-Lipschitz and $\rho$-Lipschitz, respectively.*

Assumptions 1 and 2 imply that the mapping $y^*(x)$ is $\kappa_g$-Lipschitz, where $\kappa_g = L_g/\mu > 1$ denotes the condition number of the lower level function $g$. Similarly, we denote $\kappa_f = L_f/\mu$ for the upper level function $f$ (Lin et al., 2020; Chen et al., 2021b).

Lastly, note that the problem (P) is rewritten as the regularized minimization problem $\min_{x \in \mathbb{R}^d} \Phi(x) + h(x)$, which can be nonsmooth and nonconvex. Therefore, our optimization goal is to find a critical point $x^*$ of the function $\Phi(x) + h(x)$ that satisfies the optimality condition $\mathbf{0} \in \partial(\Phi + h)(x^*)$. Here, $\partial$ denotes the following generalized notion of subdifferential.

**Definition 1.** *(Subdifferential and critical point, (Rockafellar & Wets, 2009)) The Frechét subdifferential $\widehat{\partial}F$ of a function $F$ at $x \in \text{dom } F$ is the set of $u \in \mathbb{R}^d$ defined as*

$$\widehat{\partial}F(x) = \left\{ u : \liminf_{z \neq x, z \to x} \frac{F(z) - F(x) - u^\top(z - x)}{\|z - x\|} \geq 0 \right\},$$

*and the limiting subdifferential $\partial F$ at $x \in \text{dom } F$ is the graphical closure of $\widehat{\partial}F$ defined as:*

$$\partial F(x) = \left\{ u : \exists (x_k, F(x_k)) \to (x, F(x)), \widehat{\partial}F(x_k) \ni u_k \to u \right\}.$$

*The set of critical points of $F$ is defined as $\{x : \mathbf{0} \in \partial F(x)\}$.*

## 3 Proximal Bi-level Optimization with AID

In this section, we introduce the proximal bi-level optimization algorithm with momentum accelerated approximate implicit differentiation (referred to as proximal BiO-AIDm). Recall that $\Phi(x) := f(x, y^*(x))$.

The main challenge for solving the regularized bi-level optimization problem (P) is the computation of the gradient $\nabla\Phi(x)$, which involves higher-order derivatives of the lower-level function. Fortunately, this gradient can be effectively estimated using the popular AID scheme as we elaborate below.

First, note that $\nabla\Phi(x)$ has been shown in (Ji et al., 2021) to take the following analytical form.

$$\nabla\Phi(x_k) = \nabla_x f(x_k, y^*(x_k)) - \nabla_x\nabla_y g(x_k, y^*(x_k))v_k^*,$$

where $v_k^*$ corresponds to the solution of the linear system $\nabla_y^2 g(x_k, y^*(x_k))v = \nabla_y f(x_k, y^*(x_k))$. In particular, $y^*(x_k)$ is the minimizer of the strongly convex function $g(x_k, \cdot)$, and it can be effectively approximated by running $T$ Nesterov's accelerated gradient descent updates on $g(x_k, \cdot)$ and obtaining the output $y_{k+1}$ as the approximation. With this approximated minimizer, the AID scheme estimates the gradient $\nabla\Phi(x_k)$ as follows:

$$\text{(AID):} \quad \widehat{\nabla}\Phi(x_k) = \nabla_x f(x_k, y_{k+1}) - \nabla_x\nabla_y g(x_k, y_{k+1})\widehat{v}_k^*, \tag{1}$$

where $\widehat{v}_k^*$ is the solution of the approximated linear system $\nabla_y^2 g(x_k, y_{k+1})v = \nabla_y f(x_k, y_{k+1})$, which can be efficiently solved by standard conjugate-gradient (CG) solvers. For simplicity of the discussion, we assume that $\widehat{v}_k^*$ is exactly computed throughout the paper. Moreover, the Jacobian-vector product involved in eq. (1) can be efficiently computed using the existing automatic differentiation packages (Domke, 2012; Grazzi et al., 2020).

Based on the estimated gradient $\widehat{\nabla}\Phi(x)$, we can then apply the standard proximal gradient algorithm (a.k.a. forward-backward splitting) (Lions & Mercier, 1979) to solve the regularized optimization problem (P). This algorithm is referred to as proximal BiO-AIDm and is summarized in Algorithm 1. Specifically, in each outer loop $k$, we first run $T$ accelerated gradient descent steps with Nesterov's momentum with initial point $y_k$ to minimize $g(x_k, \cdot)$ and find an approximated minimizer $y_{k+1} = y^{(T)}(x_k, y_k) \approx y^*(x_k)$, where we use the notation $y^{(T)}(x_k, y_k)$ to emphasize the dependence on $x_k$ and $y_k$. Then, this approximated minimizer is utilized by the AID scheme to estimate $\nabla\Phi(x_k)$. Finally, we apply the proximal gradient algorithm to minimize the regularized objective function $\Phi(x) + h(x)$. Here, the proximal mapping of any function $h$ at $v$ is defined as

$$\text{prox}_h(v) := \arg\min_{u\in\mathbb{R}^d}\left\{h(u) + \frac{1}{2}\|u - v\|^2\right\}.$$

---

**Algorithm 1** Proximal bi-level optimization with momentum accelerated AID (proximal BiO-AIDm)

---

1: **Input:** Stepsizes $\alpha, \beta > 0$, momentum parameter $\eta > 0$, initializations $x_0, y_0$.
2: **for** $k = 0, 1, 2, ..., K - 1$ **do**
3:     Set $y^{(0)}(x_k, y_k) = u_0 = y_k$
4:     **for** $t = 1, ...., T$ **do**
5:         $u_t = y^{(t-1)}(x_k, y_k) - \alpha\nabla_y g(x_k, y^{(t-1)}(x_k, y_k))$
6:         $y^{(t)}(x_k, y_k) = u_t + \eta(u_t - u_{t-1})$
7:     **end for**
8:     Set $y_{k+1} = y^{(T)}(x_k, y_k)$
9:     AID: compute $\widehat{\nabla}\Phi(x_k)$ according to eq. (1)
10:     Update $x_{k+1} \in \text{prox}_{\beta h}(x_k - \beta\widehat{\nabla}\Phi(x_k))$
11: **end for**
12: **Output:** $x_K, y_K$.

---

Under Assumption 1 and Assumption 2, the following lemma characterizes the smoothness of $\Phi$ and the gradient estimation error $\|\widehat{\nabla}\Phi(x_k) - \nabla\Phi(x_k)\|$ of the AID scheme.

**Lemma 1** (Ghadimi & Wang (2018)). *Let Assumptions 1.1 and 2 hold. Then, function $\Phi$ is differentiable and the gradient $\nabla\Phi$ is $L_\Phi$-Lipschitz with $L_\Phi = L_f + \frac{2L_f L_g + \tau M^2}{\mu} + \frac{\rho L_g M + L_f L_g^2 + \tau M L_g}{\mu^2} + \frac{\rho L_g^2 M}{\mu^3}$ .Moreover, the gradient estimate obtained by the AID scheme satisfies*

$$\|\widehat{\nabla}\Phi(x_k) - \nabla\Phi(x_k)\|^2 \leq \Gamma\|y_{k+1} - y^*(x_k)\|^2.$$

*where $\Gamma = 4L_f^2 + \frac{4\tau^2 M^2}{\mu^2} + \frac{4M^2\rho^2\kappa_g^2}{\mu^2} + 4L_f^2\kappa_g^2.$*

## 4 Global Convergence and Complexity of Proximal BiO-AID

In this section, we study the global convergence properties of proximal BiO-AIDm for general regularized nonconvex and nonsmooth bi-level optimization.

First, note that the main update of proximal BiO-AIDm in Algorithm 1 follows from the proximal gradient algorithm, which has been proven to generate a convergent optimization trajectory to a critical point in general nonconvex optimization (Attouch & Bolte, 2009). Hence, one may expect that proximal BiO-AIDm should share the same convergence guarantee. However, this is not obvious as the proof of convergence of the proximal gradient algorithm heavily relies on the fact that it is a descent-type algorithm, i.e., the objective function is strictly decreasing over the iterations. As a comparison, the main update of proximal BiO-AIDm applies an approximated gradient $\widehat{\nabla}\Phi(x_k)$, which is correlated with both the upper- and lower-level objective functions through the AID scheme and destroys the descent property of the proximal gradient update, and hence conceals the proof of convergence.

The following key result proves that proximal BiO-AIDm does admit an intrinsic potential function that is monotonically decreasing over the iterations. Therefore, it is indeed a descent-type algorithm, which is the first step toward establishing the global convergence.

**Proposition 1.** *Let Assumptions 1 and 2 hold and define the potential function*

$$H(x, y') := \Phi(x) + h(x) + \frac{7}{8}\|y^{(T)}(x, y') - y^*(x)\|^2. \tag{2}$$

*Choose hyperparameters $\alpha = \frac{1}{L_g}$, $\beta \leq \frac{1}{2}(L_\Phi + \Gamma + \kappa_g^2)^{-1}, \eta = \frac{\sqrt{\kappa_g}-1}{\sqrt{\kappa_g}+1}$ and $T \geq \frac{\ln(8(1+\kappa_g))}{\ln((1-\kappa_g^{-0.5})^{-1})}$. Then, the parameter sequence $\{x_k\}_k$ generated by Algorithm 1 satisfies, for all $k = 1, 2, ...,$*

$$H(x_{k+1}, y_{k+1}) \leq H(x_k, y_k) - \frac{1}{4\beta}\|x_{k+1} - x_k\|^2 - \frac{1}{8}\Big(\|y_{k+1} - y^*(x_k)\|^2 + \|y_{k+2} - y^*(x_{k+1})\|^2\Big).$$

To elaborate, the potential function $H$ consists of two components: the upper-level objective function $\Phi(x) + h(x)$ and a regularization term $\|y^{(T)}(x, y') - y^*(x)\|^2$ that tracks the optimality gap of the lower-level optimization. Hence, the potential function $H$ fully characterizes the optimization goal of the entire bi-level optimization. Intuitively, if $\{x_k\}_k$ converges to a certain critical point $x^*$ and $y^{(T)}(x_k, y_k)$ converges to $y^*(x^*)$, then it can be seen that $H(x_k, y_k)$ will converge to the local optimum $(\Phi + h)(x^*)$. To our knowledge, such a deceasing potential function has not been found for bi-level optimization in the existing literature, which is important for obtaining both the global and local convergence results later (see Theorem 1 and 2 respectively). Finding such a function is not straightforward, since the coefficient $\frac{7}{8}$ in the potential function (2) has to be elaborately selected to guarantee the monotonic decreasing property. (See the proof of Proposition 1 in Appendix A for the detail of coefficient selection.)

Based on the above characterization of the potential function, we obtain the following global convergence result of proximal BiO-AIDm in general regularized nonconvex optimization.

**Theorem 1.** *Under the same conditions as those of Proposition 1, the parameter sequence $\{x_k, y_k\}_k$ generated by Algorithm 1 satisfies the following properties.*

1. *$\|x_{k+1} - x_k\| \xrightarrow{k} 0$, $\|y_{k+1} - y^*(x_k)\| \xrightarrow{k} 0$;*

2. *The function value sequence $\{(\Phi + h)(x_k)\}_k$ converges to a finite limit $H^* > -\infty$;*

3. *The sequence $\{(x_k, y_k)\}_k$ is bounded and has a compact set of limit points. Moreover, $(\Phi + h)(x^*) \equiv H^*$ for any limit point $x^*$ of $\{x_k\}_k$;*

4. *Every limit point $x^*$ of $\{x_k\}_k$ is a critical point of the upper-level function $(\Phi + h)(x)$.*

Theorem 1 provides a comprehensive characterization of the global convergence properties of proximal BiO-AIDm in regularized nonconvex and nonsmooth bi-level optimization. Specifically, item 1 shows that the parameter sequence $\{x_k\}_k$ is asymptotically stable, and $y_{k+1}$ asymptotically converges to the corresponding minimizer $y^*(x_k)$ of the lower-level objective function $g(x_k, \cdot)$. In particular, in the unregularized case (i.e., $h = 0$), this result reduces to the existing understanding that the gradient norm $\|\nabla\Phi(x)\|$ converges to zero (Ji et al., 2021; Ghadimi & Wang, 2018; Hong et al., 2020), which does not imply the convergence of the parameter sequence. Item 2 shows that the function value sequence converges to a finite limit, which is also the limit of the potential function value sequence $\{H(x_k, y_k)\}_k$. Moreover, items 3 and 4 show that the parameter sequence $\{x_k\}_k$ converges to only critical points of the objective function, and these limit points are in a flat region where the corresponding function value is the constant $H^*$. Note that due to the nonconvexity of $\Phi$, $H^*$ is not necessarily the optimal value, i.e., it is possible that $H^* > \min_{x \in \mathbb{R}^d}(\Phi + h)(x)$. To summarize, Theorem 1 formally proves that proximal BiO-AIDm will eventually converge to critical points in nonsmooth and nonconvex bi-level optimization.

In addition to the above global convergence result, Proposition 1 can be further leveraged to characterize the computation complexity of proximal BiO-AIDm for finding a critical point in regularized nonconvex bi-level optimization. Specifically, when the regularizer $h$ is convex, we can define the following proximal gradient mapping associated with the objective function $\Phi(x) + h(x)$.

$$G(x) = \frac{1}{\beta}\Big(x - \text{prox}_{\beta h}\big(x - \beta\nabla\Phi(x)\big)\Big). \tag{3}$$

The proximal gradient mapping is a standard metric for evaluating the optimality of regularized nonconvex optimization problems (Nesterov, 2013). It can be shown that $x$ is a critical point of $\Phi(x) + h(x)$ if and only if $G(x) = \mathbf{0}$, and it reduces to the normal gradient in the unregularized case. Hence, we define the **convergence criterion** as finding a near-critical point $x$ that satisfies $\|G(x)\| \leq \epsilon$ for some pre-determined accuracy $\epsilon > 0$. We obtain the following global convergence rate and complexity of proximal BiO-AIDm.

**Corollary 1.** *Suppose $h$ is convex and the conditions of Proposition 1 hold. Then, the sequence $\{x_k\}_k$ generated by Algorithm 1 satisfies the following convergence rate.*

$$\min_{0 \leq k \leq K} \|G(x_k)\| \leq \sqrt{\frac{32}{K\beta}\big(H(x_0) - \inf_x(\Phi + h)(x)\big)}. \tag{4}$$

*Moreover, to achieve $\min_{0 \leq k \leq K} \|G(x_k)\| \leq \epsilon$, we run the algorithm with $K = 32\epsilon^{-2}(L_\Phi + \Gamma + \kappa_g^2)\big(H(x_0) - \inf_x(\Phi + h)(x)\big)$ outer iterations and $T = \frac{\ln(8(1+\kappa_g))}{\ln((1-\kappa_g^{-0.5})^{-1})}$ inner iterations, and the overall computation complexity is $KT = \frac{32\ln(8(1+\kappa_g))}{\epsilon^2\ln((1-\kappa_g^{-0.5})^{-1})}(L_\Phi + \Gamma + \kappa_g^2)\big(H(x_0) - \inf_x(\Phi + h)(x)\big)$.*

The dependence of the above computation complexity on $\epsilon$ and $\kappa := \max(\kappa_f, \kappa_g) > 1$ is no larger than $\mathcal{O}(\kappa^{3.5}(\ln\kappa)\epsilon^{-2})$. This strictly improves the computation complexity $\mathcal{O}(\kappa^4\epsilon^{-2})$ of BiO-AID that only applies to smooth nonconvex bi-level optimization (Ji et al., 2021). To elaborate the reason, in our algorithm, the $T$ Nesterov's accelerated gradient descent steps applied to $\min_y g(x_t, y)$ achieve the convergence rate $\|y_{t+1} - y^*(x_t)\| \leq (1+\kappa_g)(1-\kappa_g^{-0.5})^T\|y_t - y^*(x_t)\|$, which is faster than $\|y_{t+1} - y^*(x_t)\| \leq (1-\kappa_g^{-1})^T\|y_t - y^*(x_t)\|$ of standard gradient descent since $1-\kappa_g^{-0.5} < 1-\kappa_g^{-1}$. Therefore, to ensure that $\|y_{t+1} - y^*(x_t)\| \leq \frac{1}{4}\|y_t - y^*(x_t)\|$, Nesterov's accelerated gradient descent requires $T = \frac{\ln(8(1+\kappa_g))}{\ln((1-\kappa_g^{-0.5})^{-1})} = \mathcal{O}(\sqrt{\kappa_g}\ln\kappa_g)$ steps, which is much less than $T = \mathcal{O}(\kappa_g)$ required by standard gradient descent. On the other hand, the number of outer iterations $K$ is the same for both algorithms. Therefore, Nesterov's accelerated gradient descent yields smaller computation complexity $KT$ than that of standard gradient descent. In addition, (Ji et al., 2021) only uses smooth upper-level function $f$ while we have non-smooth regularizer $h$ which requires to analyze nonconvex proximal gradient mapping (3). To the best of our knowledge, this is the first convergence rate and complexity result of momentum accelerated algorithm for solving regularized nonsmooth and nonconvex bi-level optimization problems. We note that another momentum accelerated bi-level optimization algorithm has been studied in (Ji & Liang, 2021), which only applies to unregularized (strongly) convex bi-level optimization problems.

## 5 Convergence Rates under Local Nonconvex Geometry

In the previous section, we have proved that the optimization trajectory generated by proximal BiO-AIDm approaches a compact set of critical points. Hence, we are further motivated to exploit the local function geometry around the critical points to study its local (asymptotic) convergence guarantees, which is the focus of this section. In particular, we consider a broad class of Łojasiewicz-type geometry of nonconvex functions.

### 5.1 Local Kurdyka-Łojasiewicz Geometry

General nonconvex functions typically do not have a global geometry. However, they may have certain local geometry around the critical points that determines the local convergence rate of optimization algorithms. In particular, the Kurdyka-Łojasiewicz (KŁ) geometry characterizes a broad spectrum of local geometries of nonconvex functions (Bolte et al., 2007; 2014), and it generalizes various conventional global geometries such as the strong convexity and Polyak-Łojasiewicz geometry. Next, we formally introduce the KŁ geometry.

**Definition 2** (KŁ geometry, Bolte et al. (2014)). *A proper and lower semi-continuous function $F$ is said to have the KŁ geometry if for every compact set $\Omega \subset \mathrm{dom} F$ on which $F$ takes a constant value $F_\Omega \in \mathbb{R}$, there exist $\varepsilon, \lambda > 0$ such that for all $\bar{x} \in \Omega$ and all $x \in \{z \in \mathbb{R}^m : \mathrm{dist}_\Omega(z) < \varepsilon, F_\Omega < F(z) < F_\Omega + \lambda\}$, the following condition holds:*

$$\varphi'\left(F(x) - F_\Omega\right) \cdot \mathrm{dist}_{\partial F(x)}(\mathbf{0}) \geq 1, \tag{5}$$

*where $\varphi'$ is the derivative of $\varphi : [0, \lambda) \to \mathbb{R}_+$ that takes the form $\varphi(t) = \frac{c}{\theta} t^\theta$ for certain constant $c > 0$ and KŁ parameter $\theta \in (0, 1]$, and $\mathrm{dist}_{\partial F(x)}(\mathbf{0}) = \min_{u \in \partial F} \|u - \mathbf{0}\|$ denotes the point-to-set distance.*

As an intuitive explanation, when function $F$ is differentiable, the KŁ inequality in eq. (5) can be rewritten as $F(x) - F_\Omega \leq \mathcal{O}(\|\nabla F(x)\|^{\frac{1}{1-\theta}})$, which can be viewed as a type of local gradient dominance condition and generalizes the Polyak-Łojasiewicz (PL) condition (with parameter $\theta = \frac{1}{2}$) (Łojasiewicz, 1963; Karimi et al., 2016). In the existing literature, a large class of functions has been shown to have the local KŁ geometry, e.g., sub-analytic functions, logarithm and exponential functions and semi-algebraic functions (Bolte et al., 2014). Moreover, the KŁ geometry has been exploited to establish the convergence of many gradient-based algorithms in nonconvex optimization, e.g., gradient descent (Attouch & Bolte, 2009; Li et al., 2017), accelerated gradient method (Zhou et al., 2020), alternating minimization (Bolte et al., 2014) and distributed gradient methods (Zhou et al., 2016).

### 5.2 Convergence Rates of Proximal BiO-AIDm under KŁ Geometry

In this subsection, we obtain the following asymptotic function value convergence rates of the proximal BiO-AIDm algorithm under different parameter ranges of the KŁ geometry. Throughout, we define $k_0 \in \mathbb{N}^+$ to be a sufficiently large integer.

**Theorem 2.** *Let Assumptions 1 and 2 hold and and assume that the potential function $H$ defined in eq. (2) has KŁ geometry. Then, under the same choices of hyper-parameters as those of Proposition 1, the potential function value sequence $\{H(x_k, y_k)\}_k$ converges to its limit $H^*$ (see its definition in Theorem 1) at the following rates.*

*1. If KŁ geometry holds with $\theta \in \left(\frac{1}{2}, 1\right)$, then $H(x_k, y_k) \downarrow H^*$ super-linearly as*

$$H(x_k, y_k) - H^* \leq \mathcal{O}\left(-\left(\frac{1}{2(1-\theta)}\right)^{k-k_0}\right), \ \forall k \geq k_0; \tag{6}$$

*2. If KŁ geometry holds with $\theta = \frac{1}{2}$, then $H(x_k, y_k) \downarrow H^*$ linearly as (for some constant $C > 0$)*

$$H(x_k, y_k) - H^* \leq \mathcal{O}\left((1+C)^{-(k-k_0)}\right), \quad \forall k \geq k_0; \tag{7}$$

*3. If KŁ geometry holds with $\theta \in \left(0, \frac{1}{2}\right)$, then $H(x_k, y_k) \downarrow H^*$ sub-linearly as*

$$H(x_k, y_k) - H^* \leq \mathcal{O}\left((k-k_0)^{-\frac{1}{1-2\theta}}\right), \quad \forall k \geq k_0, \tag{8}$$

Intuitively, a larger KŁ parameter $\theta$ implies that the local geometry of the potential function $H$ is sharper, which implies an orderwise faster convergence rate as shown in Theorem 2. In particular, when the KŁ geometry holds with $\theta = \frac{1}{2}$, the proximal BiO-AIDm algorithm converges at a linear rate, which matches the convergence rate of bi-level optimization under the stronger geometry that both the upper and lower-level objective functions are strongly convex (Ghadimi & Wang, 2018). To the best of our knowledge, the above result provides the first function value converge rate analysis of proximal BiO-AIDm in the full spectrum of the nonconvex local KŁ geometry.

The proof of Theorem 2 involves two novel techniques. The first novel technique is to establish the monotonically decreasing potential function $H(x, y')$ in Proposition 1. This monotonic decreasing property guarantees that the sequence $\{(x_k, y_k)\}_k$ generated by Algorithm 1 will enter the neighborhood of critical point $(x^*, y^*(x^*))$ where the KŁ property of $H$ holds, which is essential to prove the function value convergence rates in Theorem 2. Another technical novelty is to prove the Lipschitz property of the mapping $y^{(T)}(x, y)$ in Lemma 2, which is the key to establish the asymptotic convergence rates in Theorem 2. To the best of our knowledge, this Lipschitz property has not been established in the existing literature for the mapping defined by Nesterov's accelerated gradient descent steps, and it is challenging to prove due to momentum acceleration. To address this challenge, we recursively write $y^{(t)}$ as $y^{(t)}(x, y) = (1 + \eta)G_x(y^{(t-1)}(x, y)) - \eta G_x(y^{(t-2)}(x, y))$, where $G_x(y) := y - \alpha \nabla_y g(x, y)$ is the gradient descent mapping. We then leverage the Lipschitz property of $G_x$ Hardt et al. (2016) to establish the Lipschitz property of $y^{(t)}$ via induction on $t$.

## 6 Experiment

We apply our bi-level optimization algorithm to solve a regularized data cleaning problem Shaban et al. (2019) with the MNIST dataset LeCun et al. (1998) and a linear classification model. We generate a training dataset $\mathcal{D}_{\text{tr}}$ with 20k samples, a validation dataset $\mathcal{D}_{\text{val}}$ with 5k samples, and a test dataset with 10k samples. In particular, we corrupt the training data by randomizing a proportion $p \in (0, 1)$ of their labels, and the goal of this application is to identify and avoid using these corrupted training samples. The corresponding bi-level problem is written as follows.

$$\min_\lambda \frac{1}{|\mathcal{D}_{\text{val}}|} \sum_{(x_i, y_i) \in \mathcal{D}_{\text{val}}} \Big( L\big(w^*(\lambda)^\top x_i, y_i\big) - \gamma \min(|\lambda_i|, a)\Big),$$

$$\text{where } w^*(\lambda) = \arg\min_w \frac{1}{|\mathcal{D}_{\text{tr}}|} \sum_{(x_i, y_i) \in \mathcal{D}_{\text{tr}}} \sigma(\lambda_i) L\big(w^\top x_i, y_i\big) + \rho\|w\|^2, \tag{9}$$

where $x_i, y_i$ denote the data and label of the $i$-th sample, respectively, $\sigma(\cdot)$ denotes the sigmoid function, $L$ is the cross-entropy loss, and $\rho, \gamma > 0$ are regularization hyperparameters. The regularizer $\rho\|w\|^2$ makes the lower-level objective function strongly convex. In particular, we add the nonconvex and nonsmooth regularizer $-\gamma \min(|\lambda_i|, a)$ to the upper-level objective function. (See Appendix G for the analytical solution to its proximal mapping) Intuitively, it encourages $|\lambda_i|$ to approach the large positive constant $a$ so that the training sample coefficient $\sigma(\lambda_i)$ is close to either 0 or 1 for corrupted and clean training samples, respectively. In this experiment we set $a = 20$. Therefore, such a regularized bi-level data cleaning problem belongs to the problem class considered in this paper.

We compare the performance of our proximal BiO-AIDm with several bi-level optimization algorithms, including proximal BiO-AID (without accelerated AID), BiO-AID (without accelerated AID) and BiO-AIDm (with accelerated-AID). In particular, for BiO-AID and BiO-AIDm, we apply them to solve the unregularized data cleaning problem (i.e., $\gamma = 0$). This serves as a baseline that helps understand the impact of regularization on the test performance. In addition, we also implement all these algorithms by replacing the AID scheme with the ITD scheme to demonstrate their generality.

**Hyperparameter setup.** We consider choices of corruption rates $p = 0.1, 0.2, 0.4$, regularization parameters $\gamma = 0.5, 5.0$ and $\rho = 10^{-3}$. We run each algorithm for $K = 50$ outer iterations with stepsize $\beta = 0.5$ and $T = 5$ inner gradient steps with stepsize $\alpha = 0.1$. For the algorithms with momentum accelerated AID/ITD, we set the momentum parameter $\eta = 1.0$.

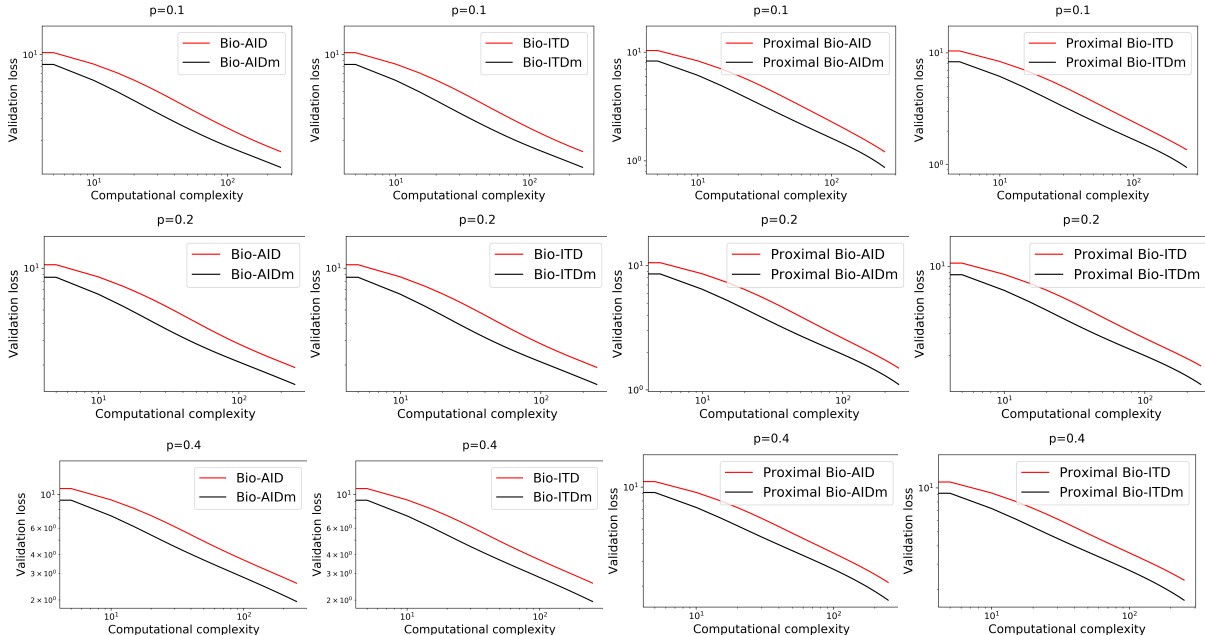

Figure 1: Comparison of bi-level optimization algorithms under data corruption rate $p = 0.1$ (top row), $p = 0.2$ (middle row) and $p = 0.4$ (bottom row). The proximal algorithms in the right 2 columns correspond to $\gamma = 5$. The $y$-axis corresponds to the upper-level objective function value, and the $x$-axis corresponds to the overall computation complexity (number of inner gradient descent steps).

## 6.1 Optimization Performance

We first investigate the effect of momentum acceleration on the optimization performance. In Figure 1, we plot the upper-level objective function value versus the computational complexity for different bi-level algorithms under different data corruption rates. In these figures, we separately compare the non-proximal algorithms and the proximal algorithms, as their upper-level objective functions are different (non-proximal algorithms are applied to solve the unregularized bi-level problem). It can be seen that all the bi-level optimization algorithms with momentum accelerated AID/ITD schemes consistently converge faster than their unaccelerated counterparts. The reason is that the momentum scheme accelerates the convergence of the inner gradient descent steps, which yields a more accurate implicit gradient and thus accelerates the convergence of the outer iterations. In addition, all the curves decrease almost in straight lines, which match the polynomial dependence of our computational complexity $\mathcal{O}(\kappa^{3.5}(\ln \kappa)\epsilon^{-2})$ on $\epsilon$ (see Corollary 1), as we plot both axes on log-scale.

## 6.2 Test Performance

To understand the impact of momentum and the nonconvex regularization on the test performance of the model, we report the test accuracy and test loss of the models trained by all the algorithms in Table 2. It can be seen that the bi-level optimization algorithms with momentum accelerated AID/ITD (columns 2 & 4 of Table 2) achieve significantly better test performance than their unaccelerated counterparts (columns 1 & 3 of Table 2). This demonstrates the advantage of introducing momentum to accelerate the AID/ITD schemes. Furthermore, we observe that the test loss decreases as the regularizer coefficient $\gamma$ increases. Therefore, adding such a regularizer improves test performance via distinguishing the sample coefficients $\sigma(\lambda_i)$ between corrupted and clean training samples. In particular, proximal BiO-AIDm and proximal BiO-ITDm with $\gamma = 5$ achieve the best test performance within each corruption rate $p$, which again demonstrates advantage of the regularizer and momentum acceleration as bolded in Table 2. Lastly, a larger corruption rate $p$ leads to a lower test performance, which is reasonable.

Table 2: Comparison of test accuracy (test loss). (Regularizer coefficient $\gamma = 0$ corresponds to four non-proximal algorithms including Bio-AID(m) and Bio-ITD(m), and $\gamma = 0.5, 5$ correspond to the proximal variants of the four algorithms. The best test accuracies and test losses of each corruption rate $p$ are bolded)

| Corruption rate | Regularizer coefficient | (Prox)Bio-AID | (Prox)Bio-AIDm | (Prox)Bio-ITD | (Prox)Bio-ITDm |
|---|---|---|---|---|---|
| $p = 0.1$ | $\gamma = 0$ | 71.00% (1.4933) | 73.80% (1.1090) | 71.10% (1.4971) | 73.80% (1.1102) |
| | $\gamma = 0.5$ | 71.00% (1.4931) | 73.80% (1.1089) | 71.10% (1.4968) | 73.80% (1.1100) |
| | $\gamma = 5$ | 71.00% (1.4914) | **73.90%** (1.1081) | 71.20% (1.4947) | **73.90% (1.1079)** |
| $p = 0.2$ | $\gamma = 0$ | 61.40% (1.7959) | 63.80% (1.3520) | 61.50% (1.8000) | 63.80% (1.3534) |
| | $\gamma = 0.5$ | 61.40% (1.7957) | 63.80% (1.3519) | 61.50% (1.7996) | 63.80% (1.3530) |
| | $\gamma = 5$ | 61.50% (1.7938) | 63.80% (1.3511) | 61.60% (1.7960) | **64.10% (1.3495)** |
| $p = 0.4$ | $\gamma = 0$ | 47.50% (2.4652) | 49.20% (1.8713) | 47.50% (2.4714) | 49.10% (1.8738) |
| | $\gamma = 0.5$ | 47.50% (2.4649) | 49.20% (1.8711) | 47.50% (2.4708) | 49.10% (1.8732) |
| | $\gamma = 5$ | 47.50% (2.4628) | 49.20% (1.8702) | 47.50% (2.4652) | **49.40% (1.8680)** |

## 7   Conclusion

In this paper, we provided a comprehensive analysis of the proximal BiO-AIDm algorithm with momentum acceleration for solving regularized nonconvex and nonsmooth bi-level optimization problems. Our key finding is that this algorithm admits an intrinsic monotonically decreasing potential function, which fully tracks the bi-level optimization progress. Based on this result, we established the first global convergence rate of proximal BiO-AIDm to a critical point in regularized nonconvex optimization, which is faster than that of BiO-AID. We also characterized the asymptotic convergence behavior and rates of the algorithm under the local KŁ geometry. We anticipate that this new analysis framework can be extended to study the convergence of other bi-level optimization algorithms, including stochastic bi-level optimization. In particular, it would be interesting to explore how bi-level optimization algorithm design affects the form of the potential function and leads to different convergence guarantees and rates in nonconvex bi-level optimization.

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

## A  Proof of Proposition 1

**Proposition 1.** *Let Assumptions 1 and 2 hold and define the potential function*

$$H(x, y') := \Phi(x) + h(x) + \frac{7}{8}\|y^{(T)}(x, y') - y^*(x)\|^2. \tag{2}$$

*Choose hyperparameters* $\alpha = \frac{1}{L_g}$, $\beta \leq \frac{1}{2}(L_\Phi + \Gamma + \kappa_g^2)^{-1}$, $\eta = \frac{\sqrt{\kappa_g}-1}{\sqrt{\kappa_g}+1}$ *and* $T \geq \frac{\ln(8(1+\kappa_g))}{\ln((1-\kappa_g^{-0.5})^{-1})}$. *Then, the parameter sequence* $\{x_k\}_k$ *generated by Algorithm 1 satisfies, for all* $k = 1, 2, ...,$

$$H(x_{k+1}, y_{k+1}) \leq H(x_k, y_k) - \frac{1}{4\beta}\|x_{k+1} - x_k\|^2 - \frac{1}{8}\Big(\|y_{k+1} - y^*(x_k)\|^2 + \|y_{k+2} - y^*(x_{k+1})\|^2\Big).$$

*Proof.* Based on the smoothness of the function $\Phi(x)$ established in Lemma 1, we have

$$\Phi(x_{k+1}) \leq \Phi(x_k) + \langle \nabla\Phi(x_k), x_{k+1} - x_k \rangle + \frac{L_\Phi}{2}\|x_{k+1} - x_k\|^2. \tag{10}$$

On the other hand, by the definition of the proximal gradient step of $x_k$, we have

$$h(x_{k+1}) + \frac{1}{2\beta}\|x_{k+1} - x_k + \beta\widehat{\nabla}\Phi(x_k)\|^2 \leq h(x_k) + \frac{1}{2\beta}\|\beta\widehat{\nabla}\Phi(x_k)\|^2, \tag{11}$$

which further simplifies to

$$h(x_{k+1}) \le h(x_k) - \frac{1}{2\beta}\|x_{k+1} - x_k\|^2 - \langle x_{k+1} - x_k, \widehat{\nabla}\Phi(x_k)\rangle. \tag{12}$$

Adding up eq. (12) and eq. (10) yields that

$$\Phi(x_{k+1}) + h(x_{k+1})$$

$$\le \Phi(x_k) + h(x_k) - \left(\frac{1}{2\beta} - \frac{L_\Phi}{2}\right)\|x_{k+1} - x_k\|^2 + \langle x_{k+1} - x_k, \nabla\Phi(x_k) - \widehat{\nabla}\Phi(x_k)\rangle$$

$$\le \Phi(x_k) + h(x_k) - \left(\frac{1}{2\beta} - \frac{L_\Phi}{2}\right)\|x_{k+1} - x_k\|^2 + \|x_{k+1} - x_k\|\|\nabla\Phi(x_k) - \widehat{\nabla}\Phi(x_k)\|$$

$$\le \Phi(x_k) + h(x_k) - \left(\frac{1}{2\beta} - \frac{L_\Phi}{2} - \frac{\Gamma}{2}\right)\|x_{k+1} - x_k\|^2 + \frac{1}{2\Gamma}\|\nabla\Phi(x_k) - \widehat{\nabla}\Phi(x_k)\|^2.$$

$$\le \Phi(x_k) + h(x_k) - \left(\frac{1}{2\beta} - \frac{L_\Phi}{2} - \frac{\Gamma}{2}\right)\|x_{k+1} - x_k\|^2 + \frac{1}{2}\|y_{k+1} - y^*(x_k)\|^2, \tag{13}$$

where the last inequality utilizes Lemma 1. Next, note that $y_{k+2} = y^{(T)}(x_{k+1}, y_{k+1})$ is generated by minimizing the strongly convex function $g$ through $T$ gradient descent steps with Nesterov's momentum with the initial point $y_{k+1}$. Hence, with $\alpha = \frac{1}{L_g}$ and $\eta = \frac{\sqrt{\kappa_g}-1}{\sqrt{\kappa_g}+1}$ (see Theorem 2.2.3 of Nesterov (2013)) , we obtain that

$$\|y_{k+2} - y^*(x_{k+1})\|^2$$

$$\le (1 + \kappa_g)(1 - \kappa_g^{-0.5})^T\|y_{k+1} - y^*(x_{k+1})\|^2$$

$$\le (1 + \kappa_g)(1 - \kappa_g^{-0.5})^T\left(2\|y_{k+1} - y^*(x_k)\|^2 + 2\|y^*(x_k) - y^*(x_{k+1})\|^2\right)$$

$$\overset{(i)}{\le} \frac{1}{4}\|y_{k+1} - y^*(x_k)\|^2 + \frac{\kappa_g^2}{4}\|x_{k+1} - x_k\|^2, \tag{14}$$

where (i) uses the fact that $y^*$ is $\kappa_g$-Lipschitz (proved in Proposition 1 of Chen et al. (2021b)) and $T \ge \frac{\ln(8(1+\kappa_g))}{\ln((1-\kappa_g^{-0.5})^{-1})}$.

Adding up eq. (14) and eq. (13) yields that

$$\Phi(x_{k+1}) + h(x_{k+1}) + \|y_{k+2} - y^*(x_{k+1})\|^2$$

$$\le \Phi(x_k) + h(x_k) - \left(\frac{1}{2\beta} - \frac{L_\Phi}{2} - \frac{\Gamma}{2} - \frac{\kappa_g^2}{4}\right)\|x_{k+1} - x_k\|^2 + \frac{3}{4}\|y_{k+1} - y^*(x_k)\|^2$$

$$\overset{(i)}{\le} \Phi(x_k) + h(x_k) - \left(\frac{1}{2\beta} - \frac{L_\Phi}{2} - \frac{\Gamma}{2} - \frac{\kappa_g^2}{4}\right)\|x_{k+1} - x_k\|^2 + \frac{3}{4}\|y_{k+1} - y^*(x_k)\|^2$$

$$\overset{(ii)}{\le} \Phi(x_k) + h(x_k) - \frac{1}{4\beta}\|x_{k+1} - x_k\|^2 + \frac{3}{4}\|y_{k+1} - y^*(x_k)\|^2$$

where (i) uses the number of iterations that $T \ge \frac{\ln 8}{\ln((1-\kappa_g^{-0.5})^{-1})}$ to ensure that $(1-\kappa_g^{-1})^T \le \frac{1}{8}$, and (ii) uses the stepsize $\beta \le \frac{1}{2}(L_\Phi + \Gamma + \kappa_g^2)^{-1}$. Defining the potential function $H(x_k, y_k) := \Phi(x_k) + h(x_k) + \frac{7}{8}\|y^{(T)}(x_k, y_k) - y^*(x_k)\|^2 = \Phi(x_k) + h(x_k) + \frac{7}{8}\|y_{k+1} - y^*(x_k)\|^2$ and rearranging the above inequality yields that

$$H(x_{k+1}, y_{k+1}) \le H(x_k, y_k) - \frac{1}{4\beta}\|x_{k+1} - x_k\|^2 - \frac{1}{8}\left(\|y_{k+1} - y^*(x_k)\|^2 + \|y_{k+2} - y^*(x_{k+1})\|^2\right).$$

□

## B   Proof of Theorem 1

**Theorem 1.** *Under the same conditions as those of Proposition 1, the parameter sequence $\{x_k, y_k\}_k$ generated by Algorithm 1 satisfies the following properties.*

1. $\|x_{k+1} - x_k\| \xrightarrow{k} 0$, $\|y_{k+1} - y^*(x_k)\| \xrightarrow{k} 0$;

2. The function value sequence $\{(\Phi + h)(x_k)\}_k$ converges to a finite limit $H^* > -\infty$;

3. The sequence $\{(x_k, y_k)\}_k$ is bounded and has a compact set of limit points. Moreover, $(\Phi + h)(x^*) \equiv H^*$ for any limit point $x^*$ of $\{x_k\}_k$;

4. Every limit point $x^*$ of $\{x_k\}_k$ is a critical point of the upper-level function $(\Phi + h)(x)$.

*Proof.* We first prove the item 1. Summing Proposition 1 from $k = 0, 1, ..., K - 1$, we obtain that for all $K \in \mathbb{N}_+$,

$$\sum_{k=0}^{K-1} \frac{1}{4\beta} \|x_{k+1} - x_k\|^2 + \frac{1}{8} \left( \|y_{k+1} - y^*(x_k)\|^2 + \|y_{k+2} - y^*(x_{k+1})\|^2 \right)$$
$$\leq H(x_0, y_0) - H(x_K, y_K)$$
$$\overset{(i)}{\leq} H(x_0, y_0) - \inf_x (\Phi + h)(x)$$
$$< +\infty. \tag{15}$$

where (i) uses $H(x, y) \geq (\Phi + h)(x)$ and the item 3 of Assumption 1 that $\Phi + h$ is lower bounded.

Letting $K \to \infty$, we further obtain that

$$\sum_{k=0}^{\infty} \frac{1}{4\beta} \|x_{k+1} - x_k\|^2 + \frac{1}{8} \left( \|y_{k+1} - y^*(x_k)\|^2 + \|y_{k+2} - y^*(x_{k+1})\|^2 \right) < +\infty. \tag{16}$$

Hence, we conclude that $\|x_{k+1} - x_k\| \to 0$, $\|y_{k+1} - y^*(x_k)\| \to 0$, which proves the item 1.

Next, we prove the item 2. We have shown in Proposition 1 that $\{H(x_k, y_k)\}_k$ is monotonically decreasing. Since $H(x_k, y_k) \geq \Phi(x_k) + g(x_k) \geq \inf_{x'} \Phi(x') + g(x')$, which is bounded below, we conclude that $\{H(x_k, y_k)\}_k$ has a finite limit $H^* > -\infty$, i.e., $\lim_{k \to \infty} (\Phi + h)(x_k) + \frac{7}{8} \|y_{k+1} - y^*(x_k)\|^2 = H^*$. Moreover, since we already showed that $\|y_{k+1} - y^*(x_k)\| \to 0$, we further conclude that $\lim_{k \to \infty} (\Phi + h)(x_k) = H^*$.

Next, we prove the item 3. $\{x_k\}_k$ is bounded since $\Phi(x_k) + g(x_k) \leq H(x_k, y_k) \leq H(x_0, y_0)$ and $\Phi + h$ has compact sub-level set. Note that

$$\|y_k\| \leq \|y_k - y^*(x_{k-1})\| + \|y^*(x_{k-1}) - y^*(0)\| + \|y^*(0)\|$$
$$\overset{(i)}{\leq} \|y_k - y^*(x_{k-1})\| + \kappa_g \|x_{k-1}\| + \|y^*(0)\|, \tag{17}$$

where (i) uses the $\kappa_g$-Lipschitz continuity of $y^*$ (Proved in Proposition 1 of Chen et al. (2021b)). Since $\|y_k - y^*(x_{k-1})\| \to 0$ and $\|x_{k-1}\|$ is bounded, the above inequality implies that $\{x_k, y_k\}_k$ is bounded and thus has compact set of limit points.

Next, we bound the subdifferential of the function. By the optimality condition of the proximal gradient update of $x_k$ and the summation rule of subdifferential, we obtain that

$$\mathbf{0} \in \partial h(x_{k+1}) + \frac{1}{\beta} \left( x_{k+1} - x_k + \beta \widehat{\nabla} \Phi(x_k) \right).$$

The above equation further implies that

$$\frac{1}{\beta} \left( x_k - x_{k+1} \right) + \nabla \Phi(x_{k+1}) - \widehat{\nabla} \Phi(x_k) \in \partial (\Phi + h)(x_{k+1}).$$

Then, we obtain that

$$\text{dist}_{\partial(\Phi+h)(x_{k+1})}(\mathbf{0}) \leq \frac{1}{\beta} \|x_{k+1} - x_k\| + \|\nabla \Phi(x_{k+1}) - \widehat{\nabla} \Phi(x_k)\|$$

$$\leq \frac{1}{\beta}\|x_k - x_{k+1}\| + \|\nabla\Phi(x_{k+1}) - \nabla\Phi(x_k)\| + \|\nabla\Phi(x_k) - \widehat{\nabla}\Phi(x_k)\|$$

$$\overset{(i)}{\leq} \left(\frac{1}{\beta} + L_\Phi\right)\|x_k - x_{k+1}\| + \sqrt{\Gamma}\|y_{k+1} - y^*(x_k)\|, \tag{18}$$

where (i) follows from Lemma 1. Since we have shown that $\|x_{k+1} - x_k\| \to 0$, $\|y_{k+1} - y^*(x_k)\| \to 0$, the above inequality implies that

$$\frac{1}{\beta}(x_k - x_{k+1}) + \nabla\Phi(x_{k+1}) - \widehat{\nabla}\Phi(x_k) \in \partial(\Phi + h)(x_{k+1}),$$

$$\text{and} \quad \frac{1}{\beta}(x_k - x_{k+1}) + \nabla\Phi(x_{k+1}) - \widehat{\nabla}\Phi(x_k) \to \mathbf{0}. \tag{19}$$

Next, consider any limit point $x^*$ of $\{x_k\}_k$ so that $x_{k(j)} \overset{j}{\to} x^*$ along a subsequence. By the proximal update of $x_{k(j)}$, we have

$$h(x_{k(j)}) + \frac{1}{2\beta}\|x_{k(j)} - x_{k(j)-1}\|^2 + \langle x_{k(j)} - x_{k(j)-1}, \widehat{\nabla}\Phi(x_{k(j)-1})\rangle$$

$$\leq h(x^*) + \frac{1}{2\beta}\|x^* - x_{k(j)-1}\|^2 + \langle x^* - x_{k(j)-1}, \widehat{\nabla}\Phi(x_{k(j)-1})\rangle.$$

Rearranging the above inequality yields that

$$h(x_{k(j)}) + \frac{1}{2\beta}\|x_{k(j)} - x_{k(j)-1}\|^2$$

$$\leq h(x^*) + \frac{1}{2\beta}\|x^* - x_{k(j)-1}\|^2 + \langle x^* - x_{k(j)}, \widehat{\nabla}\Phi(x_{k(j)-1}) - \nabla\Phi(x_{k(j)-1}) + \nabla\Phi(x_{k(j)-1})\rangle$$

$$\leq h(x^*) + \frac{1}{2\beta}\|x^* - x_{k(j)-1}\|^2 + \langle x^* - x_{k(j)}, \nabla\Phi(x_{k(j)-1})\rangle$$

$$\quad + \|x^* - x_{k(j)}\|\|\widehat{\nabla}\Phi(x_{k(j)-1}) - \nabla\Phi(x_{k(j)-1})\|$$

$$\leq h(x^*) + \frac{1}{2\beta}\|x^* - x_{k(j)-1}\|^2 + \langle x^* - x_{k(j)}, \nabla\Phi(x_{k(j)-1})\rangle$$

$$\quad + \sqrt{\Gamma}\|x^* - x_{k(j)}\|\|y_{k(j)} - y^*(x_{k(j)-1})\|.$$

Taking limsup on both sides of the above inequality and noting that $\{x_k\}_k$ is bounded, $\nabla\Phi$ is Lipschitz, $\|x_{k+1} - x_k\| \to 0$, $x_{k(j)} \overset{j}{\to} x^*$ and $\|y_{k(j)} - y^*(x_{k(j)-1})\| \overset{j}{\to} 0$, we conclude that $\limsup_j h(x_{k(j)}) \leq h(x^*)$. Since $h$ is lower-semicontinuous, we know that $\liminf_j h(x_{k(j)}) \geq h(x^*)$. Combining these two inequalities yields that $\lim_j h(x_{k(j)}) = h(x^*)$. By continuity of $\Phi$, we further conclude that $\lim_j(\Phi + h)(x_{k(j)}) = (\Phi + h)(x^*)$. Since we have shown that the entire sequence $\{(\Phi + h)(x_k)\}_k$ converges to a certain finite limit $H^*$, we conclude that $(\Phi + h)(x^*) \equiv H^*$ for all the limit points $x^*$ of $\{x_k\}_k$. This proves the item 3.

Finally, we prove the item 4. To this end, we have shown that for every subsequence $x_{k(j)} \overset{j}{\to} x^*$, we have that $(\Phi + h)(x_{k(j)}) \overset{j}{\to} H^* = (\Phi + h)(x^*)$ and there exists $u_k \in \partial(\Phi + h)(x_k)$ such that $u_k \to \mathbf{0}$ (by eq. (19)). Recall the definition of limiting subdifferential, we conclude that every limit point $x^*$ of $\{x_k\}_k$ is a critical point of $(\Phi + h)(x)$, i.e., $\mathbf{0} \in \partial(\Phi + h)(x^*)$. $\qquad\square$

## C  Proof of Corollary 1

**Corollary 1.** *Suppose $h$ is convex and the conditions of Proposition 1 hold. Then, the sequence $\{x_k\}_k$ generated by Algorithm 1 satisfies the following convergence rate.*

$$\min_{0 \leq k \leq K}\|G(x_k)\| \leq \sqrt{\frac{32}{K\beta}\big(H(x_0) - \inf_x(\Phi + h)(x)\big)}. \tag{4}$$

*Moreover, to achieve $\min_{0 \leq k \leq K} \|G(x_k)\| \leq \epsilon$, we run the algorithm with $K = 32\epsilon^{-2}(L_\Phi + \Gamma + \kappa_g^2)\big(H(x_0) -$*
*$\inf_x (\Phi + h)(x)\big)$ outer iterations and $T = \frac{\ln(8(1+\kappa_g))}{\ln((1-\kappa_g^{-0.5})^{-1})}$ inner iterations, and the overall computation*
*complexity is $KT = \frac{32\ln(8(1+\kappa_g))}{\epsilon^2 \ln((1-\kappa_g^{-0.5})^{-1})}(L_\Phi + \Gamma + \kappa_g^2)\big(H(x_0) - \inf_x(\Phi + h)(x)\big)$.*

*Proof.*

$$
\begin{aligned}
\|G(x_{k+1})\| =& \frac{1}{\beta}\|x_{k+1} - \mathrm{prox}_{\beta h}(x_{k+1} - \beta\nabla\Phi(x_{k+1}))\| \\
\overset{(i)}{\leq}& \frac{1}{\beta}\big\|x_{k+1} - x_k + \beta\big(\nabla\Phi(x_{k+1}) - \widehat{\nabla}\Phi(x_k)\big)\big\| \\
\leq& \frac{1}{\beta}\|x_{k+1} - x_k\| + \big\|\nabla\Phi(x_{k+1}) - \nabla\Phi(x_k)\big\| + \big\|\nabla\Phi(x_k) - \widehat{\nabla}\Phi(x_k)\big\| \\
\overset{(ii)}{\leq}& \Big(\frac{1}{\beta} + L_\Phi\Big)\|x_{k+1} - x_k\| + \sqrt{\Gamma}\|y_{k+1} - y^*(x_k)\| \\
\overset{(iii)}{\leq}& \frac{2}{\beta}\|x_{k+1} - x_k\| + \sqrt{\Gamma}\|y_{k+1} - y^*(x_k)\|
\end{aligned}
$$

where (i) uses $x_{k+1} \in \mathrm{prox}_{\beta h}\big(x_k - \beta\widehat{\nabla}\Phi(x_k)\big)$ and the non-expansiveness of proximal mapping since $h$ is convex, (ii) uses the property that $y^*$ is $\kappa_g$-Lipschitz continuous, and (iii) uses the stepsize $\beta \leq \frac{1}{2}(L_\Phi + \Gamma + \kappa_g^2)^{-1}$. Hence, we have

$$
\begin{aligned}
&\sum_{k=0}^{K-1} \|G(x_{k+1})\|^2 \\
&\leq 2\sum_{k=0}^{K-1}\Big(\frac{4}{\beta^2}\|x_{k+1} - x_k\|^2 + \Gamma\|y_{k+1} - y^*(x_k)\|^2\Big) \\
&\leq \max\Big(\frac{32}{\beta}, \frac{\Gamma}{16}\Big)\sum_{k=0}^{K-1}\Big(\frac{1}{4\beta}\|x_{k+1} - x_k\|^2 + \frac{1}{8}\big(\|y_{k+1} - y^*(x_k)\|^2 + \|y_{k+2} - y^*(x_{k+1})\|^2\big)\Big) \\
&\overset{(i)}{\leq} \frac{32}{\beta}\big(H(x_0) - \inf_x(\Phi + h)(x)\big),
\end{aligned}
$$

where (i) uses eq. (15) and the stepsize $\beta \leq \frac{1}{2}(L_\Phi + \Gamma + \kappa_g^2)^{-1}$ which implies that $\frac{32}{\beta} \geq \frac{\Gamma}{16}$. Hence,

$$
\min_{0 \leq k \leq K} \|G(x_k)\| \leq \sqrt{\frac{1}{K}\sum_{k=0}^{K-1}\|G(x_{t+1})\|^2} \leq \sqrt{\frac{32}{K\beta}\big(H(x_0) - \inf_x(\Phi + h)(x)\big)}.
$$

To achieve $\min_{0 \leq k \leq K} \|G(x_k)\| \leq \epsilon$, it suffices that $K \geq \frac{32}{\beta\epsilon^2}\big(H(x_0) - \inf_x(\Phi + h)(x)\big)$(the maximum possible stepsize $\beta = \frac{1}{2}(L_\Phi + \Gamma + \kappa_g^2)^{-1}$).Since each inner loop and each outer loop of Algorithm 1 involves less than 7 evaluations of gradients, Hessian-vector products and proximal mappings in total, the computational complexity is $KT = \frac{32\ln(8(1+\kappa_g))}{\epsilon^2\ln((1-\kappa_g^{-0.5})^{-1})}(L_\Phi + \Gamma + \kappa_g^2)\big(H(x_0) - \inf_x(\Phi + h)(x)\big)$. $\qquad\square$

## D  Auxiliary Lemmas for Proving Theorem 2

We first inspect the mapping $y^{(T)}(x, y)$ defined as $T$ Nesterov's accelerated gradient descent steps for minimizing $g(x, \cdot)$ with initial point $y$. Define the gradient descent operator $G_x(y) = y - \alpha\nabla_y g(x, y)$. Note that $g(x, \cdot)$ is $L_g$-smooth and $\mu$-strongly convex, and our learning rate $\alpha = \frac{1}{L_g} \leq \frac{2}{L_g + \mu}$. Hence, based on Lemma 3.6 in Hardt et al. (2016), $G_x(\cdot)$ is a contraction mapping with Lipschitz constant $1 - \frac{\alpha L_g \mu}{L_g + \mu} = \frac{\kappa_g}{\kappa_g + 1}$. Also, it can be

easily seen that $G_x(y)$ is 1-Lipschitz as a function of $x$ since $\|G_{x'}(y) - G_x(y')\| = \alpha\|\nabla g(x', y) - \nabla_y g(x, y)\| \leq \alpha L_g\|x' - x\| = \|x' - x\|$. With the operator $G_x$, the mapping $y^{(t)}$ can be recursively defined as follows.

$$y^{(0)}(x, y) = y; \tag{20}$$

$$y^{(1)}(x, y) = G_x(y); \tag{21}$$

$$y^{(t)}(x, y) = (1 + \eta)G_x(y^{(t-1)}(x, y)) - \eta G_x(y^{(t-2)}(x, y)); t \geq 2. \tag{22}$$

We can prove the above mapping $y^{(t)}$ satisfies following lemma.

**Lemma 2.** *Under Assumptions 1 & 2, $y^{(T)}(\cdot, \cdot)$ is a $(2.5^{T+1} - 1.5)$-Lipschitz continuous mapping, that is, for any two points $z := (x, y)$ and $z' := (x', y')$,*

$$\|y^{(T)}(z') - y^{(T)}(z)\| \leq (2.5^{T+1} - 1.5)\|z' - z\|.$$

*Proof.* We will prove this Lemma by induction.

Based on eq. (20), $y^{(0)}$ is 1-Lipschitz, so this Lemma holds for $T = 0$.

Based on eq. (21), the following inequality holds, which implies that this Lemma also holds for $T \geq 1$

$$\begin{aligned}\|y^1(z') - y^1(z)\| &\leq \|G_{x'}(y') - G_x(y')\| + \|G_x(y') - G_x(y)\| \\ &\leq \|x' - x\| + \frac{\kappa_g}{\kappa_g + 1}\|y' - y\| \leq \sqrt{2}\|z' - z\|\end{aligned} \tag{23}$$

Suppose this Lemma holds for any $T \leq t - 1$ $(t \geq 2)$. Then, based on eq. (22),

$$\begin{aligned}&\|y^{(t)}(z') - y^{(t)}(z)\| \\ &\leq (1 + \eta)\|G_{x'}(y^{(t-1)}(z')) - G_x(y^{(t-1)}(z))\| + \eta\|G_{x'}(y^{(t-2)}(z')) - G_x(y^{(t-2)}(z))\| \\ &\leq (1 + \eta)\|G_{x'}(y^{(t-1)}(z')) - G_x(y^{(t-1)}(z'))\| + (1 + \eta)\|G_x(y^{(t-1)}(z')) - G_x(y^{(t-1)}(z))\| \\ &\quad + \eta\|G_{x'}(y^{(t-2)}(z')) - G_x(y^{(t-2)}(z'))\| + \eta\|G_x(y^{(t-2)}(z')) - G_x(y^{(t-2)}(z))\| \\ &\leq (1 + \eta)\|x' - x\| + (1 + \eta)\frac{\kappa_g}{\kappa_g + 1}\|y^{(t-1)}(z') - y^{(t-1)}(z)\| \\ &\quad + \eta\|x' - x\| + \eta\frac{\kappa_g}{\kappa_g + 1}\|y^{(t-2)}(z') - y^{(t-2)}(z)\| \\ &\overset{(i)}{\leq} 3\|z' - z\| + 2(2.5^t - 1.5)\|z' - z\| + (2.5^{t-1} - 1.5)\|z' - z\| \\ &\leq \left(2.4(2.5^t) + 3 - 1.5\right)\|z' - z\| \leq (2.5^{t+1} - 1.5)\|z' - z\|.\end{aligned} \tag{24}$$

where (i) uses $\eta = \frac{\sqrt{\kappa_g} - 1}{\sqrt{\kappa_g} + 1} \leq 1$, $\|x' - x\| \leq \|z' - z\|$ and the assumption that $y^{(T)}(\cdot, \cdot)$ is $2.5^{T-1} - 1.5$-Lipschitz for any $T \leq t - 1$. Hence, this Lemma also holds for $T = t$ and thus for all $T \in \mathbb{N}$. $\square$

**Lemma 3.** *Under Assumptions 1 and 2, function $\|y^{(T)}(z) - y^*(x)\|^2$ is differentiable with regard to $z := (x, y)$.*

*Proof.* It suffices to prove that $y^*(x)$ and $y^{(T)}(z)$ are differentiable.

First, we prove the differentiability of $y^*(x) := \arg\min_y g(x, y)$, which satisfies the stationary condition that $\nabla_y g[x, y^*(x)] = 0$. Note that for all $x \in \mathbb{R}^d$, $\nabla_y^2 g[x, y^*(x)]$ is invertible as $g(x, \cdot)$ is strongly-convex. Therefore, the implicit function theorem implies that $y^*(x)$ is differentiable with $\nabla y^*(x) = -\left(\nabla_y^2 g[x, y^*(x)]\right)^{-1}\nabla_y\nabla_x g[x, y^*(x)]$.

Next, we will prove that $y^{(t)}(x, y)$ is differentiable for $0 \leq t \leq T$ via induction. Note that $y^{(0)}(x, y) = y$ is differentiable. Then, suppose there exists $T'$ such that $y^{(t)}$ is differentiable for all $0 \leq t \leq T' - 1$, and it suffices to prove that $y^{(T')}$ is differentiable.

Note that the gradient descent operator $G_x(y) = y - \alpha\nabla_y g(x,y)$ is differentiable with gradients

$$\nabla_1 G_x(y) := \nabla_x G_x(y) = -\alpha\nabla_x\nabla_y g(x,y), \quad \nabla_2 G_x(y) := \nabla_y G_x(y) = I - \alpha\nabla_y^2 g(x,y).$$

Therefore, based on chain rule, $y^{(t)}(x,y) = (1+\eta)G_x(y^{(t-1)}(x,y)) - \eta G_x(y^{(t-2)}(x,y))$ has the following gradients

$$\begin{aligned}
\nabla_x y^{(t)}(x,y) =& (1+\eta)\nabla_1 G_x(y^{(t-1)}(x,y)) + (1+\eta)\nabla_2 G_x(y^{(t-1)}(x,y))\nabla_x y^{(t-1)}(x,y) \\
& - \eta\nabla_1 G_x(y^{(t-2)}(x,y)) - \eta\nabla_2 G_x(y^{(t-2)}(x,y))\nabla_x y^{(t-2)}(x,y), \\
\nabla_y y^{(t)}(x,y) =& (1+\eta)\nabla_2 G_x(y^{(t-1)}(x,y))\nabla_y y^{(t-1)}(x,y) - \eta\nabla_2 G_x(y^{(t-2)}(x,y))\nabla_y y^{(t-2)}(x,y),
\end{aligned}$$

which based on the above discussion implies that $y^{(T)}$ is differentiable and thus concludes the proof. $\qquad\square$

Lemma 3 ensures that the potential function $H(x,y') := \Phi(x) + h(x) + \frac{7}{8}\|y^{(T)}(x,y') - y^*(x)\|^2$ is subdifferentiable since $\Phi + h$ is subdifferentiable. Furthermore, to prove Theorem 2 under KŁ geometry, we obtain the following bound on the subdifferential of the potential function $H$. Throughout, we denote $z := (x,y)$, $z' := (x',y')$ and $z_k := (x_k, y_k)$.

**Lemma 4.** *Let Assumptions 1 and 2 hold and consider the potential function $H$ defined in eq. (2). Then, $H$ is subdifferentiable. Furthermore, under the same choices of hyper-parameters as those of Proposition 1, the subdifferential of $H$ satisfies the following bound:*

$$\text{dist}_{\partial_z H(z_{k+1})}(\mathbf{0}) \le \frac{2}{\beta}\|x_{k+1} - x_k\| + \sqrt{\Gamma}\|y_{k+1} - y^*(x_k)\| + 2\big(2.5^{T+1} + \kappa_g\big)\|y_{k+2} - y^*(x_{k+1})\|.$$

*Proof.* Recall the potential function $H(z) := \Phi(x) + h(x) + \frac{7}{8}\|y^{(T)}(x,y) - y^*(x)\|^2$. Note that $\Phi + h$ is subdifferentiable, and $\|y^{(T)}(z) - y^*(x)\|^2$ is differentiable based on Lemma 3. Hence, by the subdifferetial rule we have

$$\begin{aligned}
\partial_z H(z) \supset & \partial_z(\Phi + h)(x) + \frac{7}{8}\partial_z(\|y^{(T)}(z) - y^*(x)\|^2) \\
= & \partial(\Phi + h)(x) \times \{\mathbf{0}\} + \frac{7}{8}\partial_z(\|y^{(T)}(z) - y^*(x)\|^2),
\end{aligned} \tag{25}$$

where the second "=" uses $\partial_y(\Phi + h)(x) = \{\mathbf{0}\}$.

Next, we derive an upper bound for the subdifferentials $\partial_z(\|y^{(T)}(z) - y^*(x)\|^2)$. Take any Frechet subdifferential $u \in \widehat{\partial}_z(\|y^{(T)}(z) - y^*(x)\|^2)$, we obtain from its definition that

$$\begin{aligned}
0 \le & \liminf_{z' \ne z, z' \to z} \frac{\|y^{(T)}(z') - y^*(x')\|^2 - \|y^{(T)}(z) - y^*(x)\|^2 - u^\top(z'-z)}{\|z'-z\|} \\
= & \liminf_{z' \ne z, z' \to z} \frac{[y^{(T)}(z') - y^*(x') + y^{(T)}(z) - y^*(x)]^\top [y^{(T)}(z') - y^*(x') - y^{(T)}(z) + y^*(x)] - u^\top(z'-z)}{\|z'-z\|} \\
\le & \liminf_{z' \ne z, z' \to z} \Bigg[ \frac{\big(\|y^{(T)}(z') - y^*(x')\| + \|y^{(T)}(z) - y^*(x)\|\big)\big(\|y^{(T)}(z') - y^{(T)}(z)\| + \|y^*(x) - y^*(x')\|\big)}{\|z'-z\|} \\
& - \frac{u^\top(z'-z)}{\|z'-z\|}\Bigg] \\
\overset{(i)}{\le} & \liminf_{z' \ne z, z' \to z} \Bigg[ \big(\|y^{(T)}(z') - y^*(x')\| + \|y^{(T)}(z) - y^*(x)\|\big)\Big(2.5^{T+1} + \kappa_g \frac{\|x'-x\|}{\|z'-z\|}\Big) - \frac{u^\top(z'-z)}{\|z'-z\|}\Bigg] \\
\overset{(ii)}{\le} & 2\big(2.5^{T+1} + \kappa_g\big)\|y^{(T)}(z) - y^*(x)\| - \limsup_{z' \ne z, z' \to z} \frac{u^\top(z'-z)}{\|z'-z\|} \\
\overset{(iii)}{=} & 2\big(2.5^{T+1} + \kappa_g\big)\|y^{(T)}(z) - y^*(x)\| - \|u\|,
\end{aligned}$$

where (i) uses Lemma 2 that $y^{(T)}(\cdot, \cdot)$ is a $(2.5^{T+1} - 1.5)$-Lipschitz continuous mapping, (ii) uses $\kappa_g \geq 1$, $2\sqrt{2} \leq 3$ and $\|x' - x\| \leq \|z' - z\|$, and the equality in (iii) is achieved by letting $z' = z + \sigma u$ with $\sigma \to 0^+$. Hence, we conclude that $\|u\| \leq 8\kappa_g \|y^{(T)}(x, y) - y^*(x)\|$. Since $\partial_z (\|y^{(T)}(z) - y^*(x)\|^2)$ is the graphical closure of $\widehat{\partial}_z (\|y^{(T)}(z) - y^*(x)\|^2)$, we have that

$$\text{dist}_{\partial_z(\|y^{(T)}(z) - y^*(x)\|^2)}(\mathbf{0}) \leq 2(2.5^{T+1} + \kappa_g)\|y^{(T)}(z) - y^*(x)\|. \tag{26}$$

Next, using the subdifferential decomposition (27), we obtain that

$$\text{dist}_{\partial_z H(z_{k+1})}(\mathbf{0})$$
$$\leq \text{dist}_{\partial(\Phi+h)(x_{k+1})}(\mathbf{0}) + \frac{7}{8}\text{dist}_{\partial_z(\|y^{(T)}(z_{k+1}) - y^*(x_{k+1})\|^2)}(\mathbf{0})$$
$$\overset{(i)}{\leq} \left(\frac{1}{\beta} + L_\Phi\right)\|x_k - x_{k+1}\| + \sqrt{\Gamma}\|y_{k+1} - y^*(x_k)\| + 2(2.5^{T+1} + \kappa_g)\|y^{(T)}(z_{k+1}) - y^*(x_{k+1})\|$$
$$\overset{(ii)}{\leq} \frac{2}{\beta}\|x_k - x_{k+1}\| + \sqrt{\Gamma}\|y_{k+1} - y^*(x_k)\| + 2(2.5^{T+1} + \kappa_g)\|y_{k+2} - y^*(x_{k+1})\|, \tag{27}$$

where (i) uses eq. (18)&(26), (ii) uses the hyperparameter choice that $\beta \leq \frac{1}{2}(L_\Phi + \Gamma + \kappa_g^2)^{-1}$ in Proposition 1. $\qquad\square$

## E  Proof of Theorem 2

**Theorem 2.** *Let Assumptions 1 and 2 hold and and assume that the potential function $H$ defined in eq. (2) has KŁ geometry. Then, under the same choices of hyper-parameters as those of Proposition 1, the potential function value sequence $\{H(x_k, y_k)\}_k$ converges to its limit $H^*$ (see its definition in Theorem 1) at the following rates.*

*1. If KŁ geometry holds with $\theta \in \left(\frac{1}{2}, 1\right)$, then $H(x_k, y_k) \downarrow H^*$ super-linearly as*

$$H(x_k, y_k) - H^* \leq \mathcal{O}\left(-\left(\frac{1}{2(1-\theta)}\right)^{k-k_0}\right), \ \forall k \geq k_0; \tag{6}$$

*2. If KŁ geometry holds with $\theta = \frac{1}{2}$, then $H(x_k, y_k) \downarrow H^*$ linearly as (for some constant $C > 0$)*

$$H(x_k, y_k) - H^* \leq \mathcal{O}\left((1 + C)^{-(k-k_0)}\right), \quad \forall k \geq k_0; \tag{7}$$

*3. If KŁ geometry holds with $\theta \in \left(0, \frac{1}{2}\right)$, then $H(x_k, y_k) \downarrow H^*$ sub-linearly as*

$$H(x_k, y_k) - H^* \leq \mathcal{O}\left((k - k_0)^{-\frac{1}{1-2\theta}}\right), \quad \forall k \geq k_0, \tag{8}$$

*Proof.* Recall that we have shown in the proof of Theorem 1 that: 1) $\{H(x_k, y_k)\}_k$ decreases monotonically to the finite limit $H^*$; 2) for any limit point $x^*$ of $\{x_k\}_k$, $H(x^*, y^*(x^*)) = (\Phi + h)(x^*)$ has the constant value $H^*$. Hence, the KŁ inequality holds after a sufficiently large number of iterations, i.e., there exists $k_0 \in \mathbb{N}^+$ such that the following holds for all $k \geq k_0$.

$$\varphi'(H(x_k, y_k) - H^*)\text{dist}_{\partial_z H(x_k, y_k)}(\mathbf{0}) \geq 1.$$

Rearranging the above inequality and utilizing eq. (27), we obtain that for all $k \geq k_0$,

$$\varphi'(H(x_k, y_k) - H^*)$$
$$\geq \frac{1}{\text{dist}_{\partial_z H(x_k, y_k)}(\mathbf{0})}$$

$$\geq \left(\frac{2}{\beta}\|x_{k-1} - x_k\| + \sqrt{\Gamma}\|y_k - y^*(x_{k-1})\| + 2\left(2.5^{T+1} + \kappa_g\right)\|y_{k+1} - y^*(x_k)\|\right)^{-1}. \tag{28}$$

For simplicity, denote $d_k := H(x_k, y_k) - H^*$ as the function value gap. Then, for a sufficiently large $k$ such that eq. (28) holds, we have

$$c^{-2}d_k^{2(1-\theta)}$$

$$\overset{(i)}{=} \left[\varphi'(d_k)\right]^{-2}$$

$$\overset{(ii)}{\leq} \left(\frac{2}{\beta}\|x_{k-1} - x_k\| + \sqrt{\Gamma}\|y_k - y^*(x_{k-1})\| + 2\left(2.5^{T+1} + \kappa_g\right)\|y_{k+1} - y^*(x_k)\|\right)^2$$

$$\overset{(iii)}{\leq} \frac{12}{\beta^2}\|x_{k-1} - x_k\|^2 + 3\Gamma\|y_k - y^*(x_{k-1})\|^2 + 24\left(5^{T+1} + \kappa_g^2\right)\|y_{k+1} - y^*(x_k)\|^2$$

$$\leq \max\left(\frac{48}{\beta}, 24\Gamma, 24(5^{T+1} + \kappa_g^2)\right)\left(\frac{1}{4\beta}\|x_{k-1} - x_k\|^2 + \frac{1}{8}\left(\|y_k - y^*(x_{k-1})\|^2 + \|y_{k+1} - y^*(x_k)\|^2\right)\right) \tag{29}$$

$$\overset{(iv)}{\leq} \max\left(\frac{48}{\beta}, 24\Gamma, 24(5^{T+1} + \kappa_g^2)\right)\left(H(x_{k-1}, y_{k-1}) - H(x_k, y_k)\right)$$

$$\leq \max\left(\frac{48}{\beta}, 24\Gamma, 24(5^{T+1} + \kappa_g^2)\right)\left(d_{k-1} - d_k\right),$$

where (i) uses the equality that $\varphi'(s) = cs^{\theta-1}$ based on Definition 2, (ii) uses eq. (28), (iii) uses the inequality that $(a + b + c)^2 \leq 3a^2 + 3b^2 + 3c^2$, and (iv) uses Proposition 1. Rearranging the above inequality yields that

$$d_{k-1} \geq d_k + Cd_k^{2(1-\theta)}, \tag{30}$$

where $C := \left[c\max\left(\frac{48}{\beta}, 24\Gamma, 24(5^{T+1} + \kappa_g^2)\right)\right]^{-1} > 0$ is a constant.

Next, we prove the convergence rates case by case.

(Case I) If $\theta \in \left(\frac{1}{2}, 1\right)$, then since $d_k \geq 0$, eq. (30) implies that $d_{k-1} \geq Cd_k^{2(1-\theta)}$, which is equivalent to that

$$C^{-\frac{1}{2\theta-1}}d_k \leq \left(C^{-\frac{1}{2\theta-1}}d_{k-1}\right)^{\frac{1}{2(1-\theta)}}.$$

Since $d_k \downarrow 0$, $C^{-\frac{1}{2\theta-1}}d_{k_0} \leq e^{-1}$ for sufficiently large $k_0 \in \mathbb{N}^+$. Hence, the above inequality implies that for $k \geq k_0$,

$$C^{-\frac{1}{2\theta-1}}d_k \leq \left(C^{-\frac{1}{2\theta-1}}d_{k_0}\right)^{\left[\frac{1}{2(1-\theta)}\right]^{k-k_0}} \leq \exp\left(-\left[\frac{1}{2(1-\theta)}\right]^{k-k_0}\right). \tag{31}$$

Since $\theta \in \left(\frac{1}{2}, 1\right)$ implies that $\frac{1}{2(1-\theta)} > 1$, the above inequality implies that $d_k \downarrow 0$ (i.e. $H(x_k, y_k) \downarrow H^*$) at the super-linear rate given by eq. (6).

(Case II) If $\theta = \frac{1}{2}$, then eq. (30) implies that

$$d_k \leq (1 + C)^{-1}d_{k-1},$$

which further implies that $d_k \downarrow 0$ (i.e. $H(x_k, y_k) \downarrow H^*$) at the linear rate given by eq. (7).

(Case III) If $\theta \in \left(0, \frac{1}{2}\right)$, then denote $\psi(s) = \frac{1}{1-2\theta}s^{-(1-2\theta)}$ and consider the following two subcases.

If $d_{k-1} \leq 2d_k$, then

$$\psi(d_k) - \psi(d_{k-1}) = \int_{d_k}^{d_{k-1}} -\psi'(s)ds = \int_{d_k}^{d_{k-1}} s^{-2(1-\theta)}ds \overset{(i)}{\geq} d_{k-1}^{-2(1-\theta)}(d_{k-1} - d_k)$$

$$\overset{(ii)}{\geq} C\left(\frac{d_k}{d_{k-1}}\right)^{2(1-\theta)} \overset{(iii)}{\geq} 2^{-2(1-\theta)}C$$

where (i) uses $d_k \leq d_{k-1}$ and $-2(1-\theta) < -1$, (ii) uses eq. (30), and (iii) uses $C > 0$, $d_{k-1} \leq 2d_k$ and $2(1-\theta) > 1$.

If $d_{k-1} > 2d_k$, then for $k \geq k_0$

$$\psi(d_k) - \psi(d_{k-1}) = \frac{1}{1-2\theta}\big(d_k^{-(1-2\theta)} - d_{k-1}^{-(1-2\theta)}\big) \overset{(i)}{\geq} \frac{1}{1-2\theta}\big(d_k^{-(1-2\theta)} - (2d_k)^{-(1-2\theta)}\big)$$

$$\geq \frac{1 - 2^{-(1-2\theta)}}{1-2\theta}d_k^{-(1-2\theta)} \overset{(ii)}{\geq} \frac{1 - 2^{-(1-2\theta)}}{1-2\theta}d_{k_0}^{-(1-2\theta)}$$

where (i) uses $d_{k-1} > 2d_k$ and $-(1-2\theta) < 0$, and (ii) uses $-(1-2\theta) < 0$, $\frac{1-2^{-(1-2\theta)}}{1-2\theta} > 0$ and $d_k \leq d_{k_0}$.

Combining the above two subcases yields that

$$\psi(d_k) - \psi(d_{k-1}) \geq \min\left[2^{-2(1-\theta)}C, \frac{1-2^{-(1-2\theta)}}{1-2\theta}d_{k_0}^{-(1-2\theta)}\right] = \frac{U}{1-2\theta} > 0, k \geq k_0$$

where $U := \min\left(2^{-2(1-\theta)}C(1-2\theta), \big(1-2^{-(1-2\theta)}\big)d_{k_0}^{-(1-2\theta)}\right) > 0$. Iterating the above inequality yields that

$$\psi(d_k) \geq \psi(d_{k_0}) + \frac{U}{1-2\theta}(k - k_0) \geq \frac{U}{1-2\theta}(k - k_0)$$

Then by substituting $\psi(s) = \frac{1}{1-2\theta}s^{-(1-2\theta)}$, the inequality above implies that that $d_k \downarrow 0$ (i.e. $H(x_k, y_k) \downarrow H^*$) at the sub-linear rate given by eq. (8). $\qquad\square$

# F   Computing Inexact Solution to the Linear System $\nabla_y^2 g(x_k, y_{k+1})v = \nabla_y f(x_k, y_{k+1})$.

In the approximate gradient $\widehat{\nabla}\Phi(x)$ defined in (1), we assume access to the exact solution $\widehat{v}_k^*$ of the approximated linear system $\nabla_y^2 g(x_k, y_{k+1})v = \nabla_y f(x_k, y_{k+1})$ for simplicity. In this section, we will consider using standard conjugate-gradient (CG) solvers to obtain an inexact solution, and prove that such inexactness almost does not increase the order of computation complexity.

Denote $\widetilde{v}_k^*$ as the inexact solution obtained by $N$ iterations of CG with initialization 0. Then, the approximation error of CG can be derived as follows.

$$\|\widetilde{v}_k^* - \widehat{v}_k^*\| \overset{(i)}{\leq} 2\sqrt{\kappa_g}\Big(\frac{\sqrt{\kappa_g}-1}{\sqrt{\kappa_g}+1}\Big)^N\|\widehat{v}_k^*\| \overset{(ii)}{\leq} \frac{2M\sqrt{\kappa_g}}{\mu}(1-\kappa_g^{-1/2})^N, \tag{32}$$

where (i) uses eq. (17) of Grazzi et al. (2020) with initialization 0 and (ii) uses $\|\widehat{v}_k^*\| = \|[\nabla_y^2 g(x_k, y_{k+1})]^{-1}\nabla_y f(x_k, y_{k+1})\| \leq \frac{M}{\mu}$ (since $g(x, \cdot)$ is $\mu$-strongly convex and $f$ is $M$-Lipscithz.)

Then, replacing the exact solution $\widehat{v}_k^*$ with the inexact solution $\widetilde{v}_k^*$ in the approximate gradient (1) we define the new approximate gradient and update rule as follows

$$\widetilde{\nabla}\Phi(x_k) = \nabla_x f(x_k, y_{k+1}) - \nabla_x \nabla_y g(x_k, y_{k+1})\widetilde{v}_k^*. \tag{33}$$

The approximation error of the above approximate gradient has the following upper bound.

$$\|\widetilde{\nabla}\Phi(x_k) - \nabla\Phi(x_k)\| \leq \|\widetilde{\nabla}\Phi(x_k) - \widehat{\nabla}\Phi(x_k)\| + \|\widehat{\nabla}\Phi(x_k) - \nabla\Phi(x_k)\|$$

$$\overset{(i)}{\leq} \|\nabla_x \nabla_y g(x_k, y_{k+1})\|\|\widetilde{v}_k^* - \widehat{v}_k^*\| + \Gamma\|y_{k+1} - y^*(x_k)\|^2$$

$$\overset{(ii)}{\leq} 2M\kappa_g^{1.5}(1-\kappa_g^{-1/2})^N + \Gamma\|y_{k+1} - y^*(x_k)\|^2,$$

where (i) uses eqs. (1) & (33) and Lemma 1, and (ii) uses eq. (32) and $\|\nabla_x \nabla_y g(x_k, y_{k+1})\| \leq L_g$ (since $\nabla g(z)$ is $L_g$-smooth). Compared with the gradient error bound in Lemma 1, the above bound has the additional

term $2M\kappa_g^{1.5}(1 - \kappa_g^{-1/2})^N$. Hence, using $\widetilde{\nabla}\Phi$ instead of $\Phi$ in the proof of Proposition 1, $2M\kappa_g^{1.5}(1 - \kappa_g^{-1/2})^N$ will be added to eq. (13), i.e.,

$$\Phi(x_{k+1}) + h(x_{k+1})$$
$$\leq \Phi(x_k) + h(x_k) - \left(\frac{1}{2\beta} - \frac{L_\Phi}{2} - \frac{\Gamma}{2}\right)\|x_{k+1} - x_k\|^2 + \frac{1}{2\Gamma}\|\nabla\Phi(x_k) - \widetilde{\nabla}\Phi(x_k)\|^2.$$
$$\leq \Phi(x_k) + h(x_k) - \left(\frac{1}{2\beta} - \frac{L_\Phi}{2} - \frac{\Gamma}{2}\right)\|x_{k+1} - x_k\|^2 + \frac{1}{2}\|y_{k+1} - y^*(x_k)\|^2 + M\Gamma^{-1}\kappa_g^{1.5}(1 - \kappa_g^{-1/2})^N.$$

Adding the above inequality to eq. (14), we have

$$\Phi(x_{k+1}) + h(x_{k+1}) + \|y_{k+2} - y^*(x_{k+1})\|^2$$
$$\leq \Phi(x_k) + h(x_k) - \frac{1}{4\beta}\|x_{k+1} - x_k\|^2 + \frac{3}{4}\|y_{k+1} - y^*(x_k)\|^2 + M\Gamma^{-1}\kappa_g^{1.5}(1 - \kappa_g^{-1/2})^N,$$

which implies that

$$H(x_{k+1}, y_{k+1}) \leq H(x_k, y_k) - \frac{1}{4\beta}\|x_{k+1} - x_k\|^2 - \frac{1}{8}\left(\|y_{k+1} - y^*(x_k)\|^2 + \|y_{k+2} - y^*(x_{k+1})\|^2\right)$$
$$+ M\Gamma^{-1}\kappa_g^{1.5}(1 - \kappa_g^{-1/2})^N.$$

Telescoping the above inequality, we obtain that

$$\sum_{k=0}^{K-1} \frac{1}{4\beta}\|x_{k+1} - x_k\|^2 + \frac{1}{8}\left(\|y_{k+1} - y^*(x_k)\|^2 + \|y_{k+2} - y^*(x_{k+1})\|^2\right)$$
$$\leq H(x_0, y_0) - \inf_x(\Phi + h)(x) + KM\Gamma^{-1}\kappa_g^{1.5}(1 - \kappa_g^{-1/2})^N, \tag{34}$$

which is a slight modification of eq. (15) with the additional term $KM\Gamma^{-1}\kappa_g^{1.5}(1 - \kappa_g^{-1/2})^N$. Therefore, near the end of the proof of Corollary 1 in Appendix C, replacing eq. (15) with eq. (34), we obtain the following new convergence rate

$$\min_{0 \leq k \leq K} \|G(x_k)\| \leq \sqrt{\frac{32}{K\beta}\left(H(x_0) - \inf_x(\Phi + h)(x)\right) + M\Gamma^{-1}\kappa_g^{1.5}(1 - \kappa_g^{-1/2})^N}.$$

To achieve $\min_{0 \leq k \leq K} \|G(x_k)\| \leq \epsilon$, we simply replace the number of outer iterations $K \geq \frac{32}{\beta\epsilon^2}\left(H(x_0) - \inf_x(\Phi + h)(x)\right) = \mathcal{O}(\kappa_g^2(\kappa_f + \kappa_g)\epsilon^{-2})$ with $K \geq \frac{64}{\beta\epsilon^2}\left(H(x_0) - \inf_x(\Phi + h)(x)\right) = \mathcal{O}(\kappa_g^2(\kappa_f + \kappa_g)\epsilon^{-2})$, and set $N \geq \frac{\ln(2M(\Gamma\epsilon)^{-1}\kappa_g^{1.5})}{-\ln(1 - \kappa_g^{-1/2})} = \mathcal{O}(\sqrt{\kappa_g}\ln(\kappa_f\kappa_g\epsilon^{-1}))$ (Note that $\Gamma = \mathcal{O}(\kappa_f\kappa_g^2)$ as shown in Proposition 1). All the other hyperparameter choices are not changed. Since there are $K$ outer iterations and each outer iteration contains $T$ inner gradient descent steps for minimizing $g(x, \cdot)$ and $N$ CG steps, the overall computation complexity is $K(T + N) = \mathcal{O}(\kappa_g^{2.5}(\kappa_f + \kappa_g)\ln(\kappa_f\kappa_g\epsilon^{-1})\epsilon^{-2})$, which has almost the same order as the one $\mathcal{O}(\kappa_g^{2.5}(\kappa_f + \kappa_g)(\ln\kappa_g)\epsilon^{-2})$ in Corollary 1 with difference of logarithm level.

## G  The Proximal Mapping of the Regularizer in the Experiment

All the AID-type and ITD-type algorithms implemented in the experiment require computing the following proximal mapping with stepsize $\beta > 0$.

$$\text{prox}_{\beta h}(v) := \arg\min_{u \in \mathbb{R}^d}\left\{h(u) + \frac{1}{2\beta}\|u - v\|^2\right\}.$$

In the experiment in Section 6, we use the non-smooth and nonconvex regularizer $h(\lambda) = -\frac{\gamma}{|\mathcal{D}_{\text{val}}|}\sum_{(x_i, y_i) \in \mathcal{D}_{\text{val}}} \min(|\lambda_i|, a)$, for which the $i$-th entry of the above proximal mapping $[\text{prox}_{\beta h}(\lambda)]_i$ has analytical solution in the following two cases.

(Case 1) If $a > \frac{\gamma}{|\mathcal{D}_{\mathrm{val}}|}$, then

$$[\mathrm{prox}_{\beta h}(\lambda)]_i := \begin{cases} -a; -a < \lambda_i < -\left(a - \frac{\gamma}{|\mathcal{D}_{\mathrm{val}}|}\right) \\ \lambda_i - \frac{\gamma}{|\mathcal{D}_{\mathrm{val}}|}; -\left(a - \frac{\gamma}{|\mathcal{D}_{\mathrm{val}}|}\right) \leq \lambda_i < 0 \\ \lambda_i + \frac{\gamma}{|\mathcal{D}_{\mathrm{val}}|}; 0 \leq \lambda_i < a - \frac{\gamma}{|\mathcal{D}_{\mathrm{val}}|} \\ a; a - \frac{\gamma}{|\mathcal{D}_{\mathrm{val}}|} \leq \lambda_i < a \\ \lambda_i; |\lambda_i| \geq a \end{cases}.$$

(Case 2) If $a \leq \frac{\gamma}{|\mathcal{D}_{\mathrm{val}}|}$, then

$$[\mathrm{prox}_{\beta h}(\lambda)]_i := \begin{cases} -a; -a < \lambda_i < 0 \\ a; 0 \leq \lambda_i < a \\ \lambda_i; |\lambda_i| \geq a \end{cases}.$$

