# OpenReview forum: "A Fast and Convergent Proximal Algorithm for Regularized Nonconvex and Nonsmooth Bi-level Optimization"
_TMLR — Rejected by TMLR_

### Review · Reviewer_BwHr · 2022-08-05

**Summary Of Contributions:**

This paper proposes a proximal BiO-AIDm method for solving the bilevel optimization problem. The bilevel optimization problem has smooth loss function with nonsmooth and nonconvex regularization term, and the lower-level objective is assumed to be strongly convex. Using the accelerated gradient method to solve the lower objective function inexactly, the approximate gradient of upper-level function can be obtained. Then, the proximal (inexact-)gradient method is extended to solve the problem. Global convergence to stationary points is proved and local convergence rate is presented under Kurdyka-Łojasiewicz conditions.

**Broader Impact Concerns:**

Not applied.

**Requested Changes:**

I have the following comments to improve the presentation of the paper.
1. Log-scale plots will better show the convergence rate in experiment section.
2. The proximal mapping of $-\gamma \min( |\lambda_i|, a)$ should be given.
3. In page 5, ' Throughout the paper, we denote $\kappa=L/\mu$ as the condition number of the bi-level problem (P)' might be removed, since the condition number has been defined in page 4.
4. In Assumption 2, why do you assume that   $\nabla f, \nabla g$ have the same Lipschitz constant? I think using different notations may reflect the effects of different functions in the convergence results.
5. A table list of convergence results for BiO-AID under different settings will be helpful.

**Strengths And Weaknesses:**

Strengths:
1. Under the assumption that nonsmooth regularization $h$ is convex, the computations complexity of the proposed method is $\mathcal{O}(\kappa^{3.5}/\epsilon^2)$, which is better than known results in (Ji et al., 2021).
2.  The numerical experiments show that the proposed momentum based methods improve the efficiency and test accuracy.


Weakness:
1. The notation $y^T(x_k,y_k)$ is ambiguous although  $\top$ represents the transpose in the paper.
2. Many typos : $\Phi+g$ which should be $\Phi+h$, see equation (4), and proof of Corollary 1.
3. A notation is missing in experiment section: where $w^*(\lambda)=\arg min ...$

---

> ### Author Response · Authors · 2022-08-30
> **Authors' reply to Reviewer BwHr**
>
> Thank you very much for reviewing our manuscript and providing valuable feedback. Below is a response to the review comments. We have submitted a revised version with all revisions marked in "red". Please let us know if further clarifications are needed.
>
> **Q1:** $y^T(x_k,y_k)$ may be misinterpreted as transpose.
>
> **A:** Thank you for your great point. We have changed to $y^{(T)}(x_k,y_k)$ in the revision.
>
> **Q2:** $\Phi+g$ should be $\Phi+h$.
>
> **A:** Thank you for pointing that out. We have corrected that in the revision.
>
> **Q3:** $\mathop{\arg\min}_w$ is missing in the experiment section.
>
> **A:** Thank you for pointing that out. We have corrected that in the revision.
>
> **Q4:** Log-scale plots will better show the convergence rate in experiment section.
>
> **A:** Thank you for your great suggestion, we have changed to log-scale in the revision.
>
> **Q5:** The proximal mapping of $-\gamma\min(|\lambda_i|,a)$ should be given.
>
> **A:** Thank you. We have added that to Appendix G in the revision.
>
> **Q6:** In page 5, 'Throughout the paper, we denote $\kappa=L/\mu$ as the condition number of the bi-level problem (P)' might be removed, since the condition number has been defined in page 4.
>
> **A:** Thank you for pointing that out. We have removed that sentence in the revision.
>
> **Q7:** In Assumption 2, why do you assume that $\nabla f$, $\nabla g$ have the same Lipschitz constant? I think using different notations may reflect the effects of different functions in the convergence results.
>
> **A:** Thank you for your great suggestion. We have changed to different Lipschitz constants and accordingly adjusted all the equations, theoretical results and proofs, etc.
>
> **Q8:** A table list of convergence results for BiO-AID under different settings will be helpful.
>
> **A:** Thank you for your great suggestion. We have added this table in the revision.

---

### Review · Reviewer_UknX · 2022-08-12

**Summary Of Contributions:**

This paper develops a new algorithm for solving a bilevel optimization problem where the upper level problem is nonconvex (the sum of a smooth component and a nonsmooth regularizer) and the lower level problem is strongly convex. The algorithmic techniques are (i) approximate implicit differentiation (AID) (ii) momentum for accelerating the computation of implicit gradients used in AID and (iii) proximal gradient. The theoretical results of the method are three-fold: (i) In the general setting sketched above, the authors prove the subsequential convergence of the iterate to a critical point of the solution (ii) when the regularizer is additionally assumed to be convex, then the complexity of finding a point with $\epsilon$-small prox-gradient mapping is $O(\kappa^{3.5}\epsilon^{-2})$. This complexity, when specialized to the un-regularized case studied in the literature, improves the existing complexity by a factor of $\kappa^{1/2}$. (iii) when a Lojasiewicz-type assumption holds, the authors show asymptotic rates for the algorithm.


**Broader Impact Concerns:**

I don't have concerns on the ethical implications of the work, so I don't think adding a Broader Impact Statement is necessary.

**Requested Changes:**

- When does Assumption 3 hold? Is it assumed in any of the literature? Does this assumption hold for the practical experiments in the paper?

- What are the insights that allows the authors to improve the rates of Ji et al., 2021 in the particular case of non-regularized problem? Please also see my 2nd comment in the "Weaknesses" part above.

- What is the advantage of momentum? Does it give a faster rate such as Nesterov's acceleration does? What kind of momentum is referred to in the paper? The subroutine in Algorithm 1 looks like the FISTA version of acceleration. Explaining this would be useful.

- Beginning of page 6, the authors talk about the difficulties arising with bilevel optimization. Can the authors clarify if similar issues occur in the non-regularized case, say in the setting of Ji et al. 2021? If that is the case, are the authors using some ideas from Ji et al., 2021? It is perfectly okay if this is the case, being clear is good for the reader to appreciate the novelties in the paper better.

- What are the cases when prox operator of a nonsmooth and nonconvex regularizer is tractable? Is this the case for the regularizer used in the experiments?

- end of page 4, the authors say for simplicity they focus on the case when $v_k^*$ is exactly computed. Can they show the effect of inexactness to convince the reader this restriction is without loss of generality?

- After Assumption 1, the authors say Assumption 1 is satisfied for regularizers used in practice. What are some examples? Is this assumption satisfied for the regularizer considered in the experiments of the paper?

- When stating problem (P), it is not yet clear why the argmin set of the lower level problem is a singleton (this becomes clear later as strong convexity is assumed in the lower problem). It might be nice to clarify this by either making this an inclusion or adding to the sentence after (P) that strong convexity is assumed for the lower level problem.

- Theorem 2 is a little confusing. I could not find $H^*$ defined properly, I only see it in Theorem 1. Theorem 1 suggests that $H^*$ is the limit that the sequence in Theorem 1. (2) converges to, but not necessarily the optimal value. Is this the same $H^*$ in Theorem 2? So the rate in Theorem 2 is to the value of the limit but not necessarily the optimal value? This is not clear since with PL, sometimes we can show global optimality, i.e., convergence to a globally optimal objective value. From what I understand, this is not the case here, but this should be explained.

- The notation "global convergence" is used several times in the paper. This is a little confusing since in the context of nonconvex optimization, "global" sometimes refer to convergence to the global optimum. I think the authors use this term to say "convergence independent of initialization", but I think this should be made clearer.

- Where exactly is the analysis for Step 9 in Algorithm 1, AID? The authors sometimes refer to accelerated AID, what does this mean? Does this mean AID with the implicit gradients computed with the subroutine in the Algorithm 1 which uses momentum? Or is there any other acceleration mechanism going on? If so, where in the analysis can we see this?

- Table 1 is confusing, please consider using bold fonts for the best results so the reader can easily see. The authors say increasing regularization gives better test loss, but I cannot see this in the table. Which entries does the authors' comment correspond to? Please consider reformatting this table and improving the explanation: for example, the authors can add which lines they refer to in the table when explaining the improvements.

- For eq. (13), the authors cite Nesterov's book, please give a precise pointer. It is not clear where in the book the reader needs to look at to see what the authors are using for this inequality.

**Strengths And Weaknesses:**

I think the paper satisfies the criteria of the journal, namely bilevel optimization is definitely of interest to the ML community and the claims in the paper are supported by theoretical analysis and empirical comparisons.

Strengths:

- The paper generalizes the existing guarantees for gradient based nonconvex bilevel optimization. In particular, the paper adds a regularizer to the objective that can be nonconvex and nonsmooth. For the results in Theorem 1 (asymptotic iterate convergence), the regularizer can be nonsmooth and nonconvex. For Corollary 2 (complexity for prox gradient mapping), the regularizer needs to be additionally convex. An additional assumption (Assumption 3) is required for getting the asymptotic rates with Lojasiewicz-type assumption. In all cases, regularizer is accessed with a prox operator.

- Also very importantly, the paper improves the existing complexities when specialized to a particular case. In particular, when there is no regularizer in the problem, the paper improves the existing SOTA of Ji et al., 2021 by a factor of $\kappa^{1/2}$ where $\kappa$ is the condition number. This is a desirable property: the paper has a general analysis but also improves in particular cases.

- The rates with Lojasiewicz-type assumption seem to be new.

Weaknesses: My main concerns about the lack of explanation about the assumptions/insights and notation at times throughout the paper. For example

- Assumption 3 is not explained. When does this assumption hold? Is it assumed in any of the literature? Does this assumption hold for the practical experiments in the paper?

- The main insight that allows the authors to improve the rates of Ji et al., 2021 in the particular case of non-regularized problem is not clear. This is important for the authors to clarify the main technical contributions of the paper. For example, the authors can be more clear on what kind of tools are used from the existing literature such as Ji et al., 2021 and what are the new tools that were required to improve the complexity and generalize the results.

- The mechanisms of using momentum and its benefits are not explained.

- The presentation of the empirical results is unclear, Table 1 should be improved and explained better.

---

> ### Author Response · Authors · 2022-08-30
> **Authors' reply to Reviewer UknX (1)**
>
> Thank you very much for reviewing our manuscript and providing valuable feedback. Below is a response to the review comments. We have submitted a revised version with all revisions marked in "red". Please let us know if further clarifications are needed.
>
> **Q1:** When does Assumption 3 hold? Is it assumed in any of the literature? Does this assumption hold for the practical experiments in the paper?
>
> **A:** Great questions. A similar assumption was made in [1] to analyze the asymptotic convergence of GDA algorithm in minimax optimization. There, they assumed
> that the function $m(x):=\|y^*(x)-y\|^2$ is subdifferentiable for any $y$.
>
> We later found that Assumption 3 can actually be proved under Assumptions 1 and 2. Therefore, in Appendix D of the revision, we turned Assumption 3 into Lemma 3 and added its proof. Note that the proof only uses the following properties of the lower level function $g$: $\nabla_x\nabla_y g$, $\nabla_y^2 g$ exist and $g(x,\cdot)$ is strongly convex for all $x\in\mathbb{R}^d$. The function $g$ in our experiment satisfy these properties, so $\omega^*(\lambda)$ satisfies the new Lemma 3.
>
> References:
>
> [1] Chen, Z., Zhou, Y., Xu, T., \& Liang, Y. (2021). Proximal gradient descent-ascent: Variable convergence under KL geometry. In Proc. International Conference on Learning Representations (ICLR).
>
> **Q2:** What tools are used from the existing literature such as Ji et al., 2021? What are the new tools that were required to improve the complexity and generalize the results?
>
> **A:** Great questions. The existing techniques that we exploited include Lemma 1 (basic properties of function $\Phi$) and standard convergence rate of Nesterov's accelerated gradient algorithm.
>
> There are two major technical novelties as elaborated in the the revision (including Section 1.1, after Proposition 1 and after Theorem 2). The first novelty is to establish a monotonically decreasing potential function $H(x,y'):=\Phi(x)+h(x)+\frac{7}{8}\|y^*(x,y')-y^*(x)\|$ in Proposition 1 such that $H(x_{k+1}, y_{k+1}) < H(x_k,y_k)$, which is critical for proving both the global (non-asymptotic) convergence rate in Section 4 and the local (asymptotic) convergence rate in Section 5. In particular, the decreasing property of the potential function guarantees that the algorithm will eventually stay within a local region and then one can expect to prove a local convergence rate.
> In contrast, [2] did not establish such a potential function and they establish only the global convergence rate.
> Another technical novelty is to prove the Lipschitz property of the mapping $y^T(x,y)$ in Lemma 2, which is defined by applying $T$ Nesterov’s accelerated gradient descent steps to minimize $g(x,\cdot)$ with initialization $y$. This property is the key to establish the asymptotic convergence rate in Section 5. To the best of our knowledge, this Lipschitz property has not been established in the existing literature for the mapping defined by Nesterov’s accelerated gradient descent steps, and it is challenging to prove due to momentum acceleration. To address this challenge, we recursively write $y^t$ as $y^{t}(x,y)=(1+\eta)G_x(y^{t-1}(x,y))-\eta G_x(y^{t-2}(x,y))$, where $G_x$ is the gradient descent mapping. We then leverage the Lipschitz property of $G_x$ to establish the Lipschitz property of $y^t$ via induction on $t$.
>
> [2] Ji, K., Yang, J., \& Liang, Y. (2021, July). Bilevel optimization: Convergence analysis and enhanced design. In International Conference on Machine Learning (pp. 4882-4892). PMLR.
>
> **Q3:** The mechanisms of using momentum and its benefits are not explained.
>
> **A:** Thanks for the suggestion. We have added more elaboration on that right after Corollary 1 and we also summarize it here. The $T$ Nesterov's accelerated gradient descent steps applied to $\min_y g(x_t,y)$ achieve the convergence rate $\|y_{t+1}-y^*(x_t)\|\le(1+\kappa)(1-\kappa^{-0.5})^T \|y_t-y^*(x_t)\|$, which is faster than $\|y_{t+1}-y^*(x_t)\|\le(1-\kappa^{-1})^T \|y_t-y^*(x_t)\|$ of standard gradient descent since $1-\kappa^{-0.5}<1-\kappa^{-1}$. Therefore, to ensure that $\|y_{t+1}-y^*(x_t)\|\le \frac{1}{4} \|y_t-y^*(x_t)\|$, Nesterov's accelerated gradient descent requires $T=\mathcal{O}(\sqrt{\kappa}\ln\kappa)$ steps, which is much less than $T=\mathcal{O}(\kappa)$ required by standard gradient descent. On the other hand, the number of outer iterations $K$ is the same for both algorithms. Therefore, Nesterov's accelerated gradient descent yields smaller computation complexity $KT$ than that of standard gradient descent.
>
> **Q4:** The presentation of the empirical results is unclear. Table 1 should be improved and explained better.
>
> **A:** Thank you for pointing that out. Please see the answer to Q15 for the improvements that we have made in the revision.

---

> > ### Author Response · Authors · 2022-08-30
> > **Authors' reply to Reviewer UknX (2)**
> >
> > **Q5:** What are the insights that allows the authors to improve the rates of Ji et al., 2021 in the particular case of non-regularized problem?
> >
> > **A:** In the inner loop of the algorithm, Ji et al., 2021 uses standard gradient descent while we use Nesterov's accelerated gradient descent. We have elaborated the reason why the latter algorithm leads to a lower computation complexity in the answer to Q3.
> >
> > **Q6:** What is the advantage of momentum? Does it give a faster rate such as Nesterov's acceleration does? What kind of momentum is referred to in the paper? The subroutine in Algorithm 1 looks like the FISTA version of acceleration.
> >
> > **A:** Great questions. Momentum accelerates the convergence rate of the inner gradient descent steps, as elaborated in the answer to Q3 and also right after Corollary 1 in the revision.
> >
> > In the paper, we apply gradient descent with constant Nesterov's momentum to solve the unregularized subproblem $\min_y g(x,y)$ involved in the inner loop of our algorithm (the regularizer $h(x)$ is addressed by the proximal mapping in the outer loop). In comprison, FISTA is designed for regularized problems and it adopts a special diminishing momentum coefficient.
> >
> > **Q7:** Does non-regularized bilevel optimization in Ji et al., 2021 also have the difficulties stated in page 6? Have we used some ideas from Ji et al., 2021?
> >
> > **A:** Great questions. The difficulty of only having access to the approximated gradient $\widehat{\nabla}\Phi(x_k)\approx\nabla\Phi(x_k)$ also exists in Ji et al. 2021, and this is natural as both works adopt the AID scheme to approximate the gradient. Also, both our work and Ji et al. 2021 obtain similar one-step progress of the outer gradient descent step following standard nonconvex convergence analysis (see eq. (13)). Here, one major difference is that we have a possibly nonconvex regularizer $h(x)$ and thus need to analyze nonconvex proximal gradient updates using subdifferential, while Ji et al. 2021 does not consider any regularizer.
> >
> > Moreover, in the later part of the convergence proof, Ji et al. 2021 iterates eq. (14) over $k$ and then substitute it into eq. (13), which is used for obtaining only global convergence rate of $\|\nabla \Phi(x_t)\|$. In comparison, we directly substitute eq. (14) into eq. (13) to construct a non-increasing potential function, which is the key to obtain not only the global convergence rate of $\|\nabla \Phi(x_t)\|$ in Theorem 1, but also the asymptotic function value convergence rates under K\L~geometry in Theorem 2.
> >
> > In the revision, we do not claim the novelty of using the approximation $\widehat{\nabla}\Phi(x_k)\approx\nabla\Phi(x_k)$, as it is also adopted in Ji et al. 2021. Instead, we added the above two novelties in comparison with existing literature (not only Ji et al. 2021). The novelty in nonconvex regularizer is added after Corollary 1, and the novelty in constructing non-increasing potential function is added in Section 1.1, after Proposition 1 and after Theorem 2.

---

> > > ### Author Response · Authors · 2022-08-30
> > > **Authors' reply to Reviewer UknX (3)**
> > >
> > > **Q8:** What are the cases when proximal operator of a nonsmooth and nonconvex regularizer is tractable? Is this the case for the regularizer used in the experiments?
> > >
> > > **A:** Great questions. We list the following three examples of nonsmooth and nonconvex regularizers with tractable proximal operator, including the regularizer used in our experiment (see Example 3).
> > >
> > > (Example 1) $h(x)=\lambda||x|| _ 0$ is an $\ell_0$-norm regularizer where $\lambda>0$ and $||x|| _ 0$ denotes the number of nonzero entries of vector $x$. The $i$-th entry of the proximal operator $[\text{prox} _ {\beta h}(x)] _ i$ has the following analytical solution:
> > > $[\text{prox} _ { \beta h }] _ i = {x_i}; |x_i|>\sqrt{2\lambda}$
> > >
> > > $[\text{prox} _ { \beta h }] _ i = 0; |x_i|\le\sqrt{2\lambda}$
> > >
> > > (Example 2) $h(X):=\sum _ {i=1}^m \widehat{r}(\sigma_i)$ is a low rank regularizer of matrix $X$ where $X=U\text{diag}(\sigma_1,\ldots,\sigma_m)V$ ($\sigma_1\ge\ldots\ge\sigma_m\ge 0$) is the SVD of $X$ and $\widehat{r}(\sigma)$ is a concave and
> > > non-decreasing function of $\sigma\ge 0$ with $\widehat{r}(0)=0$ [3]. Then $\text{prox} _ {\lambda h}(X)=U\text{diag}(y^*)V$ where the $i$-th entry of $y^*$ is given by $y_i^*\in \arg\min _ {y_{i} \geq 0} \frac{1}{2}(y_{i}-\sigma_{i})^{2}+\mu \widehat{r}(y_{i})$, which has tractable analytical solution for the five examples of $\widehat{r}$ listed in Table I of [3].
> > >
> > > [3] Yao, Q., Kwok, J. T., \& Zhong, W. (2015, November). Fast low-rank matrix learning with nonconvex regularization. In 2015 IEEE International conference on data mining (pp. 539-548). IEEE.
> > >
> > > (Example 3) For the regularizer $ h( \lambda ):=-\frac{\gamma}{|\mathcal{D} _ {\text{val}}|} \sum _ {(x_i,y_i)\in \mathcal{D}_{\text{val}}} \min(|\lambda_i|,a)$ used in our experiment, the $i$-th entry of the proximal operator $[\text{prox} _ {\beta h}(\lambda)]_i$ has tractable analytical solution as shown in Appendix G of the revision.
> > >
> > > **Q9:** Can we show the effect of inexactness of $v_k^*$ to convince the reader this restriction is without loss of generality?
> > >
> > > **A:** Great suggestion. We have added a discussion on the effect of inexactness of $v_k^*$ in Appendix F, and found that using conjugate gradient algorithm to compute an inexact $v_k^*$ only increases the order of the overall computation complexity by certain logarithm factors.
> > >
> > > **Q10:** What regularizers satisfy the item 2 of Assumption 1, i.e., they are proper and lower-semicontinuous (possibly nonsmooth and nonconvex)? Does the regularizer in the experiment satisfy this assumption?
> > >
> > > **A:** Great questions. Examples of proper and lower-semicontinuous regularizers include any proper convex functions (can be nonsmooth, e.g., $\ell_1$ norm), $\ell_p$ norm with $p>0$ (can be nonconvex and nonsmooth), and all the three examples mentioned in the answer to Q8 (nonsmooth and nonconvex).
> > >
> > > **Q11:** We can either use $y^*(x)\in\mathop{\arg\min}_{y\in\mathbb{R} ^ {p}}~g(x,y)$ in the problem (P) or state strong convexity assumption after the problem (P).
> > >
> > > **A:** Thank you for your great suggestion. We have used $y^*(x)\in\mathop{\arg\min}_{y \in \mathbb{R} ^ p }~g(x,y)$ in the revision.
> > >
> > > **Q12:** Do Theorems 1 and 2 share the same $H^*$? Is $H^*$ optimal?
> > >
> > > **A:** Great questions. Theorems 1 and 2 share the same $H^*$, which is the limit of both $\Phi(x_t)+g(x_t)$ and $H(x_t,y_t)$ and is not necessarily the global optimum. We have clarified this in Theorem 2 and after Theorem 1. Thank you for pointing that out.
> > >
> > > **Q13:** Reconsider 'global convergence'.
> > >
> > > **A:** Great suggestion. We use 'global convergence' to differentiate the non-asymptotic convergence results with arbitrary initialization under **global** nonconvexity geometry from the asymptotic results under **local** Kurdyka-{\L}ojasiewicz geometry. We have clarified the definition of 'global convergence' and 'local convergence' at their first appearance in Section 1.1 and the beginning of Section 5 respectively. Thank you for pointing that out.
> > >
> > > **Q14:** Where exactly is the analysis for Step 9 in Algorithm 1, AID? What does accelerated AID mean? Is there any other acceleration mechanism?
> > >
> > > **A:** Step 9 in Algorithm 1 computes $\widehat{\nabla}\Phi(x_t)\approx\nabla\Phi(x_t)$. The analysis of its approximation error $||\widehat{\nabla}\Phi(x_t)-\nabla\Phi(x_t)||$ is given by Lemma 1, which has been proved in Ghadimi \& Wang (2018).
> > > Accelerated AID means AID with momentum acceleration used in the subroutine. We did not use any other acceleration techniques.
> > >
> > > **Q15:** In the table of experimental result, use bold fonts for the best results. Reformat table and indicate the lines when explaining the improvements. For example, from which entries can we see that increasing $\gamma$ gives better test loss?
> > >
> > > **A:** Thank you for your great suggestions. In the revision, we have reformatted the table and improved the explanation.

---

> > > > ### Author Response · Authors · 2022-08-30
> > > > **Authors' reply to Reviewer UknX (4)**
> > > >
> > > > **Q16:** Where in Nesterov (2014) can we find the convergence rate of Nesterov's momentum used in eq. (13) (now eq. (14) in the revision)?
> > > >
> > > > **A:** Great question. It is in Theorem 2.2.3 of [4]. We have also added this exact location right above eq. (14) in the revision.
> > > >
> > > > [4] Yurii Nesterov. Introductory lectures on convex optimization: A basic course, volume 87.
> > > > Springer Science \& Business Media, 2013.

---

### Review · Reviewer_GbVW · 2022-08-16

**Summary Of Contributions:**

The paper presents an algorithm for bilevel optimization problem where the inner problem is assumed to be strongly convex.

The algorithm is of a proximal gradient type and has two loops. At each iteration of the outer loop we run a proximal gradient update where the gradient is computed by running an accelerated gradient descent in the inner loop.

I think the results presented in the paper are quite expected and are not due to a new technique or any new ideas, but simply to strong assumptions. The main workhorse is Lemma 1 which was obtained in [Ghadimi & Wang, 2018] and the rest is rather straightforward derivations.

Below I give a more specific critique.

**Broader Impact Concerns:**

-

**Requested Changes:**

Please address carefully each point above.

**Strengths And Weaknesses:**

### Strong assumptions
Bounded level sets is a restrictive assumption taking into account that we have a non-convex problem.
    In assumption 2 almost everything is assumed to be Lipschitz: outer objective $f$,  both gradients $\nabla f$, $\nabla g$, and Hessians/Jacobians $\nabla^2_yg$, $\nabla_x\nabla_yg$. Not only that we don't have much of examples that satisfy them, but also the constants are unlikely to be available which are needed for deriving the proposed complexity.

### Complexity
Obtained complexity is confusing. First of all, it is not clear how the authors use notation $a = \mathcal{O}(b)$, when both $a$ and $b$ are constants. Secondly, I am not sure why the dependence between all the Lipschitz constants is the one that authors suggested to provide us $\mathcal{O}(\kappa^3)$ in Lemma 1. I don't see any connections between $\tau $ and $M$, for instance.

### Convergence issues
I do not understand how the authors proved convergence of the sequence $y_k$. This sequence is the output of $T$ steps of accelerated GD applied to a strongly convex inner problem. However, we don't increase $T$ but take it as a constant. For instance, even if we start from a solution $x_0=x^*$, after $T$ iterations we obtain some approximations of the true $y^*$ and I don't see how we can get closer without increasing $T$.

### Minor
1.    There is no much discussion about the mathematics behind the algorithm. What was novel/difficult there?
2.   I don't see the point of using the word "fast" in the title. There is nothing fast about the algorithm. The same with "convergent". Readers should infer this from the word "algorithm" anyway. In continuous optimization, algorithms without analysis are usually called heuristics.

---

> ### Author Response · Authors · 2022-08-30
> **Authors' reply to Reviewer GbVW (1)**
>
> Thank you very much for reviewing our manuscript and providing valuable feedback. Below is a response to the review comments. We have submitted a revised version with all revisions marked in "red". Please let us know if further clarifications are needed.
>
> **Q1:** The assumptions on bounded sub-level set and various Lipschitz properties are too strong.
>
> **A:** Thanks for the comment. We think these assumptions can be justified or relaxed. Specifically, for the bounded sub-level set assumption, a continuous function $f(x)$ satisfies this assumption if it is coercive, i.e., $\lim_{\|x\|\to \infty} f(x) = +\infty$. Hence, if the regularizer $h$ is coercive (e.g., $\ell_p$ norm with $p>0$) and $\Phi$ is bounded, then $\Phi+h$ is guaranteed to have bounded sub-level sets.
>
> Regarding the global Lipschitz assumptions. In fact, they can be relaxed to local Lipschitz conditions along the optimization trajectory $\{x_t, y_t\}_t$. In fact, item 3 of Theorem 1 implies a bounded trajectory $\{x_t, y_t\}_t$, and therefore the local Lipschitz constants are guaranteed to exist. Technically, both the global Lipschitz assumptions and local Lipschitz assumptions lead to the same proof steps, and we just need to use the corresponding Lipschitz constants.
>
> **Q2:** What does $a=\mathcal{O}(b)$ mean for constants $a, b$? How is $\mathcal{O}(\kappa^3)$ obtained in Lemma 1 without $\tau$ and $M$?
>
> **A:** Good question. $a=\mathcal{O}(b)$ precisely means that there exists a constant $C>0$ which does not depend on $\kappa=L_1/\mu$ such that $a\le Cb$. In the paper, we also use the notation $a=\Omega(b)$ to indicate that $a\ge Cb$ for some constant $C>0$ that does not depend on $\kappa$. These notations are standard.
>
> Regarding the $\mathcal{O}(\kappa^3)$ mentioned in Lemma 1, we follow [1] and only consider the dependence on the condition number $\kappa$, which is usually much larger than the Lipschitz constants and other constants for ill-conditioned problems. Therefore, the other constants such as $M$, $\tau$, $\rho$ are absorbed into the $\mathcal{O}()$ notation. We have clarified this in the revision. Thank you for pointing this out.
>
> [1] Ji, K., Yang, J., \& Liang, Y. (2021, July). Bilevel optimization: Convergence analysis and enhanced design. In International Conference on Machine Learning (pp. 4882-4892). PMLR.
>
> **Q3:** How can we prove convergence of $y_k$ with constant $T$?
>
> **A:** Great question. We set a fixed inner iteration number $T=\mathcal{O}(\sqrt{\kappa}\ln\kappa)$, which ensures that $\|y_{k+1}-y^*(x_k)\|\le \frac{1}{4}\|y_k-y^*(x_k)\|$ at the $k$-th outer iteration. This guarantees the key inequality eq. (13) to hold, as shown below.
> \begin{align}
> 	\|y_{k+2}-y^*(x_{k+1})\|^2 &\le \frac{1}{4} \|y_{k+1}-y^*(x_k)\|^2 + \frac{\kappa^2}{4}\|x_{k+1}-x_k\|^2.  \quad\quad\quad(14)\nonumber
> \end{align}
> We note that at the initial outer iterations (i.e., when $k$ is small), $\|x_{k+1}-x_k\|$ is relatively large and so is $\|y_{k+2}-y^*(x_{k+1})\|$. However, as $x_k$ converges, $\|x_{k+1}-x_k\|$ is close to zero so that based on eq. (14), $\|y_{k+1}-y^*(x_k)\|$ will decay almost exponentially fast with regard to outer iteration $k$. For more technical details, please refer to the proof of Proposition 1 in Appendix A.

---

> > ### Author Response · Authors · 2022-08-30
> > **Authors' reply to Reviewer GbVW (2)**
> >
> > **Q4:** What's the novelty/difficulty in the mathematics behind the algorithm?
> >
> > **A:** Great question. Our paper establishes the first accelerated convergence result on regularized nonsmooth and nonconvex bi-level optimization. Regarding the novelty of algorithm design, our algorithm has both proximal mapping in the outer loop and momentum acceleration in the inner loop. We have mentioned these two designs right above Algorithm 1.
> >
> > Regarding the technical novelties in the analysis. The major novelty is to establish a monotonically decreasing potential function $H(x,y'):=\Phi(x)+h(x)+\frac{7}{8}\|y^*(x,y')-y^*(x)\|$ in Proposition 1 such that $H(x_{k+1}, y_{k+1}) < H(x_k,y_k)$, which is critical for proving both the global (non-asymptotic) convergence rate in Section 4 and the local (asymptotic) convergence rate in Section 5. In particular, the decreasing property of the potential function guarantees that the algorithm will eventually stay within a local region and then one can expect to prove a local convergence rate. In contrast, [1] did not establish such a potential function and they establish only the global convergence rate.
> >
> > Another technical novelty is to prove the Lipschitz property of the mapping $y^T(x,y)$ in Lemma 2, which is defined by applying $T$ Nesterov’s accelerated gradient descent steps to minimize $g(x,\cdot)$ with initialization $y$. This property is the key to establish the asymptotic convergence rate in Section 5. To the best of our knowledge, this Lipschitz property has not been established in the existing literature for the mapping defined by Nesterov’s accelerated gradient descent steps, and it is challenging to prove due to momentum acceleration. To address this challenge, we recursively write $y^t$ as $y^{t}(x,y)=(1+\eta)G_x(y^{t-1}(x,y))-\eta G_x(y^{t-2}(x,y))$, where $G_x$ is the gradient descent mapping. We then leverage the Lipschitz property of $G_x$ to establish the Lipschitz property of $y^t$ via induction on $t$. We have elaborated on both novelties in the revision (including Section 1.1, after Proposition 1 and after Theorem 2).
> >
> > [1] Ji, K., Yang, J., \& Liang, Y. (2021, July). Bilevel optimization: Convergence analysis and enhanced design. In International Conference on Machine Learning (pp. 4882-4892). PMLR.
> >
> > **Q5:** What's the point of using "fast" and "convergent" in the title?
> >
> > **A:** Great suggestion. In the revision, We have removed 'convergent' and changed 'fast' to 'accelerated' since we adopted Nesterov's acceleration scheme, which helps improve our algorithm's computation complexity by a factor of $\mathcal{O}(\kappa^{0.5})$.

---

### Decision · Action_Editors · 2022-09-26

**Recommendation:** Reject

**Comment:**

In this paper, the authors consider the nonconvex bi-level optimization problem with possibly nonconvex and nonsmooth regularizers. They develop a proximal BiO-AIDm algorithm, which utilizes Nesterov’s momentum to accelerate the computation of the implicit gradient involved in AID. Then, the authors analyze the convergence properties of their algorithm, under different conditions.

This paper is well-motivated, and the authors successfully establish an improved computation complexity over existing results. However, the reviewer has serious concerns (even after rebuttal) regarding the assumption and the complexity. In particular, the assumption on bounded sub-level sets is too strong for non-convex problems, and the hiding constants in the final complexity could be very large. Furthermore, the algorithmic novelty is limited, as the idea of using Nesterov’s momentum in the inner loop is quite straightforward. Another limitation is that there is no lower bound, and thus we do not know whether the complexity is tight.